# In vitro HIV DNA integration in *STAT3* drives T cell persistence—A model of HIV-associated T cell lymphoma

Michael Rist[1], Machika Kaku[1¤], John M. Coffin [1,2]*

**1** Graduate Program in Immunology, Tufts University, Boston, Massachusetts, United States of America, **2** Department of Molecular Biology and Microbiology, Tufts University, Boston, Massachusetts, United States of America

¤ Current address: Department of Medical Microbiology and Immunology, University of Wisconsin, Madison, Wisconsin, United States of America
* John.coffin@tufts.edu

## Abstract

Oncogenic retroviruses are known for their pathogenesis via insertional mutagenesis, in which the presence of a provirus and its transcriptional control elements alter the expression of a nearby or surrounding host gene. There are reports of proviral integration driving oncogenesis in people with HIV and the use of HIV-derived vectors for gene therapy has raised concern about oncogenic side effects. To study this issue, we used an in vitro primary human CD4 + T cell infection model developed in our laboratory to identify HIV-1 integration sites that might influence cell proliferation or survival. Combining integration site analysis and bulk RNA sequencing, we found that an upregulated *STAT3* signature due to proviral insertional mutagenesis was associated with persistent HIV-infected CD4 + T cells. HIV+ persistent cells also expressed a STAT3-related anti-apoptotic and cytotoxic phenotype that resembles that of HIV-associated T cell lymphomas. HIV insertional mutagenesis of *STAT3* and expression of its downstream targets provides a model of HIV-associated T cell lymphomas that can be used to further determine the oncogenic drivers of HIV-associated lymphomas, both AIDS- and gene therapy-associated, and, potentially, to evaluate therapeutics against these HIV-associated cancers.

## Author summary

The effects of HIV proviral insertional mutagenesis have been demonstrated in a handful of HIV-associated T cell lymphomas, where integration of an HIV provirus within intron 1 of *STAT3*, results in increased expression of the STAT3 protein. To study the effects of HIV insertional mutagenesis, we established an in vitro culture protocol of primary human CD4 + T cells infected with a replication-incompetent HIV vector with a gfp reporter.

**Data availability statement:** All relevant data are within the paper and its Supporting information files.

**Funding:** This research was supported by Research Grants R35 CA 200421 from the National Cancer Institute (www.cancer.gov) and RO1 AI 184043 from the Institute for Allergy and Infectious Disease (www.niaid.nih.gov) to JMC. The funders had no role in study design, data collection and analysis, decision to publish, or preparation of the manuscript.

**Competing interests:** I have read the journal's policy, and the authors of this manuscript have the following com-peting interests: MR is a minority shareholder and former employee of Sana Biotechnology, Inc. JMC was a mem-ber of the Scientific Advisory Board and a shareholder of ROME Therapeutics, Inc and the Scientific Advisory Board and a shareholder of Generate Biomedicines, Inc. MK has no competing interests.

After infection, the HIV/GFP+ cells from three donors declined, but, over time, 3/6 replicates from one donor populations of infected cells rebounded. The resurgent HIV/GFP+ cells contained a provirus integrated within intron 1 of *STAT3*, which led to increases in gene expression, *STAT3* activation, and up-regulation of a *STAT3*-associated anti-apoptotic and cytotoxic phenotype. The *STAT3*-associated gene signature shared similarities to the HIV-associated lym-phomas with similar integration sites. Additionally, in all 3 replicates, insertional mutagenesis of genes other than *STAT3* may have also contributed to clonal expansion of HIV/GFP+T cells.

Overall, we have demonstrated that HIV provirus insertional mutagenesis can influence T cell persistence. Our study provides a primary T cell culture model system that can be used to further study how proviral insertional mutagenesis influences HIV-associated T cell lymphomas and the safety of lentiviral vectors used in gene and cell therapies.

## Introduction

A necessary step in retrovirus replication is the integration of DNA produced by reverse transcription of the virion RNA into the host genome to form a provirus. In well-studied animal models, integration of a retroviral provirus can lead to insertional misexpression of proto-oncogenes driving cancer development [1–3]. In humans, there is extensive evidence that gammaretroviral vectors used for gene therapy can ultimately lead to altered cell proliferation and oncogenesis [4–7]. Enhancer elements of the long terminal repeats (LTRs) retained in these vectors used as a possible treatment for severe combined immunodeficiency (SCID)-X1, and Wiskott-Aldrich Syndrome (WAS), led to insertional mutagenesis of host genes, particularly LMO2, causing increased levels of transcription and protein [4–7]. Given the safety con-cerns, many gene therapies and modified cell therapies have switched to modified HIV-based vector systems with self-inactivating LTRs, commercially known as "lenti-viral vectors." Despite there being a handful of approved cell and gene therapies that use lentiviral vectors for gene insertion, the FDA has issued a warning concerning the development of secondary T cell malignancies associated with these therapies [8,9]. While the benefits of these therapies outweigh the risk, it is still important to under-stand the role HIV-modified vectors can play in the increased risk [10–15].

 As with other retroviruses, there is clear, albeit limited, evidence that HIV-1 infec-tion can lead to oncogenesis through insertional mutagenesis [16,17], although most cancers associated with HIV-1 infection are due to the effects of immunodeficiency, allowing for replication of opportunistic oncogenic viruses or, possibly, loss of immu-nosurveillance leading to unchecked tumor growth [3,18,19]. Treatment with antiret-roviral therapy (ART) is an effective tool for preventing progression to AIDS in people with HIV (PWH) [19]. However, PWH are still at an increased risk of non-Hodgkins lymphomas (NHL), often of B cell origin, even while adhering to ART [20]. Continued risk of NHL has been thought to be triggered by ongoing immune dysregulation and

expression of HIV proteins by PWH on ART [21–23]; However, there is one published example of an AIDS-associated B cell lymphoma with a clonal integration of an HIV provirus in intron 1 of *STAT3* [16], a gene whose misexpression is commonly associated with non-HIV-associated lymphomas [24]. Further examples of proviral insertion in this common B cell malignancy have not been reported.

In addition to the known drivers of HIV-associated cancer, there is evidence that HIV-mediated insertional mutagenesis can contribute to relatively rare T cell lymphomas [17]. In particular, two AIDS-associated high-grade T cell lymphoma samples were also shown to contain clonal proviruses integrated in the first intron of *STAT3.* Analysis of chimeric LTR-*STAT3* transcripts in these lymphomas showed that the 3' HIV LTR was driving overexpression of *STAT3*. Additionally, several anaplastic large cell lymphomas (ALCL), of T cell origin, from PWH were found to have clonal proviral integrations in both *STAT3* and the first intron of the *src* family tyrosine kinase *LCK*. Within all the lymphomas, the protein product of the LTR-promoted allele of *STAT3* was in its active phosphorylated state [17].

*STAT3* is an important transcription factor, which plays a variety of roles in lymphocyte function and differentiation, and which can also influence tumor development [25–28]. *STAT3* signaling pathways influence immune cells proliferation, response to infection, and differentiation into memory phenotype for future responses [27]. *STAT3* gain-of-function mutations are often associated with oncogenesis by playing an active role in promoting abnormal cell proliferation or survival [26]. *STAT3* dysregulation has been reported to play significant roles in NHL including those of both B and T cell origin [29–31]. Increased *STAT3* activity leads to upregulation of its downstream targets including well characterized protooncogenes such as *MYC* and *BCL2* [26,29]. *STAT3* mutations account for about 20% of ALCL development, often associated with additional mutations [31]. Therefore, HIV proviral promoter insertion driven *STAT3* expression could be an oncogenic event contributing to increased risk of NHL of T cell origin.

While examples of HIV proviral integration driving oncogenesis may be limited to a handful of cases of PWH, integration within oncogenes is quite common for HIV-1. HIV-1 preferentially infects actively dividing cells and integrates into highly expressed genes, many of which can also be classified as oncogenes [32–36]. Integration of defective proviruses within one of seven genes is associated with increased clonal expansion or survival of infected cells, likely resulting from insertional mutagenesis [37,38]. However, *STAT3* was not one of the seven oncogenes identified as such targets in non-cancer cells in vivo. This evidence suggests that integration can provide cells with a growth or survival advantage even if it does not lead to oncogenesis. However, integration within these seven oncogenes only plays a minor role in maintenance of the HIV reservoir in vivo [38]. The most important – perhaps the only – factor driving clonal expansion is specific antigen recognition and stimulation of infected persistent memory CD4 + T cells [38–40], which contribute to the maintenance of the viral reservoir [41,42]. Overall, there are several factors contributing to HIV+ persistent cells and therefore multiple elements can work in combination to drive oncogenesis or persistence of HIV+ cells.

In a previous study [43], our group developed an in vitro system to monitor promotion of clonal expansion of infected primary CD4 + T cells by a replication-defective HIV vector. Although we found no evidence for selective growth of cells with proviruses in any of the seven oncogenes implicated in the in vivo studies [38], we did observe reproducible selection for cells with proviruses integrated into *STAT3* introns 1–3.

Based on this result, we hypothesized that in vitro HIV-1 proviral integration within *STAT3* in primary human T cells can model HIV-driven survival as a precursor to oncogenesis. In the present study, we used a new integration site analysis (ISA) pipeline to characterize the role of *STAT3* integration in primary human CD4 + T cells infected in vitro. Preferentially expanded HIV+ clonal populations with *STAT3* integration sites showed increased expression and activation of STAT3, leading to a STAT3-associated increase in anti-apoptotic and cytotoxic factors, expression signatures similar to ALCL lymphomas. However, only one of three replicate infected cultures showed integration in *STAT3* as the sole driver of HIV/GFP+ cell survival advantage, whereas two other replicates had additional proviral integrations within other genes (*DCUN1D4*, *LOC100310756*, or *CCDC77*) that may have contributed to HIV/GFP+ cell survival. The proviral integration-driven *STAT3* dysregulation contributed to both an anti-apoptotic and cytotoxic phenotype, which contributed to

in vitro HIV + T cell persistence in culture. Most importantly, our model system can provide a similar environment to the in vivo development of NHL of T cell origin in PWH, as well, potentially, as development of oncogenic side-effects of lentiviral vector-mediated gene therapy. This system of selection for advantageous effects of proviral insertional mutagenesis will also allow for further study of additional mutations that may be needed for the development of T cell lymphomas.

## Results

Our laboratory has previously reported the development of a culture model of in vitro HIV infection of primary human CD4 + T cells to assess the role of HIV proviral integration in HIV + T cell persistence [43]. This study identified sites of integration of a replication-incompetent HIV vector leading to clonal expansion of infected cells in six replicate cultures from two separate donors (Donor 1 and Donor 2). The cells were stimulated by anti-CD3/CD28 beads prior to transduction and restimulated on days 12 and 42 after transduction. Clonally expanded cells were identified on day 57 post-infection. Integration site analysis (ISA) [44] identified a cluster of integration sites, mostly in intron 3 of STAT3, with multiple breakpoints implying preferential growth and/or survival of the host cell. Additionally, insertional mutagenesis of STAT3 driven by the HIV proviral LTR was confirmed through detection of spliced HIV-STAT3 chimeric transcripts initiated in the 5' LTR [37,45,46]. The current study was designed to expand upon these findings and better characterize the contribution of insertional mutagenesis of STAT3 to HIV + T cell clonal expansion by assessing STAT3 and STAT3-related genes at a transcriptional and protein level and compare in vitro infected cells to T cell lymphomas with similar integration sites from PWH.

### HIV integration provides a survival advantage to in vitro infected primary human CD4 + T cells

Since proviral insertional mutagenesis of STAT3 has been associated with clonal expansion, we first wanted to see if HIV integration is associated with T cell persistence over a longer period in culture. Frozen aliquots of primary human CD4 + CD45RA + T cells from three different HIV-negative donors (Donors 3, 4, and 5) were obtained from STEMCELL Technologies Inc. and thawed in RPMI medium containing IL-2. No live participants were included in this study. After 4 days of stimulation with anti-CD3/CD28 beads, each donor sample was split into replicates, with Donor 3 and 5 being split into three replicates each and Donor 4 into six replicates. The activated T cells were transduced with a replication incompetent, VSV G-protein pseudotyped HIV vector, with a gfp reporter located in place of nef [47], at an MOI of ~1 gfp transducing-unit per cell (TU/cell). T cells from each donor were cultured for extended periods until there were no longer enough viable cells for collection (Fig 1A). IL-2 was added every three days and anti-CD3/CD28 beads added just after the cultures reached maximum expansion, as their number began to decline. Cultures were split as necessary and a portion at each split was cryopreserved for future analysis. Throughout the time course, cells from the different donors and replicates were assessed by flow cytometry for Gfp expression as a reporter for HIV integration and expression.

Over time following infection, the percentage of Gfp-positive (GFP+) cells in each culture declined, an observation consistent with prior work by others [47] but differing from our previous experience using similar culturing methods [43]. This decline supports the idea that production of viral proteins or Gfp is harmful to infected T cells [48,49] or that the HIV provirus was being silenced over time [50]. In three of the six replicates from Donor 4, a resurgence in Gfp expression was detected starting around day 80, and two of those replicates surpassed 50% of the cells expressing Gfp (and therefore were HIV+) on day 126, indicating a growth advantage in the infected relative to uninfected cells (Fig 1B).

To test whether GFP+ cell reduction was due to proviral silencing, qPCR was performed to measure the number of HIV proviruses per cell. ACH2 cells were used as a standard as this cell line has been described to have close to 1 provirus per cell despite having a low level of ongoing viral replication which leads to amplification in highly passaged cultures [51–53]. A standard curve of one HIV vector plasmid per one genome of DNA was used to determine the number of HIV proviruses in our ACH2 cell DNA. Primers were targeted to the HIV LTR to detect proviral DNA and amplification was normalized to CCR5, a host gene with two copies per diploid genome in primary cells [54]. We independently determined that our ACH2 cell DNA contained about 2 proviruses per cell (S1 Fig).

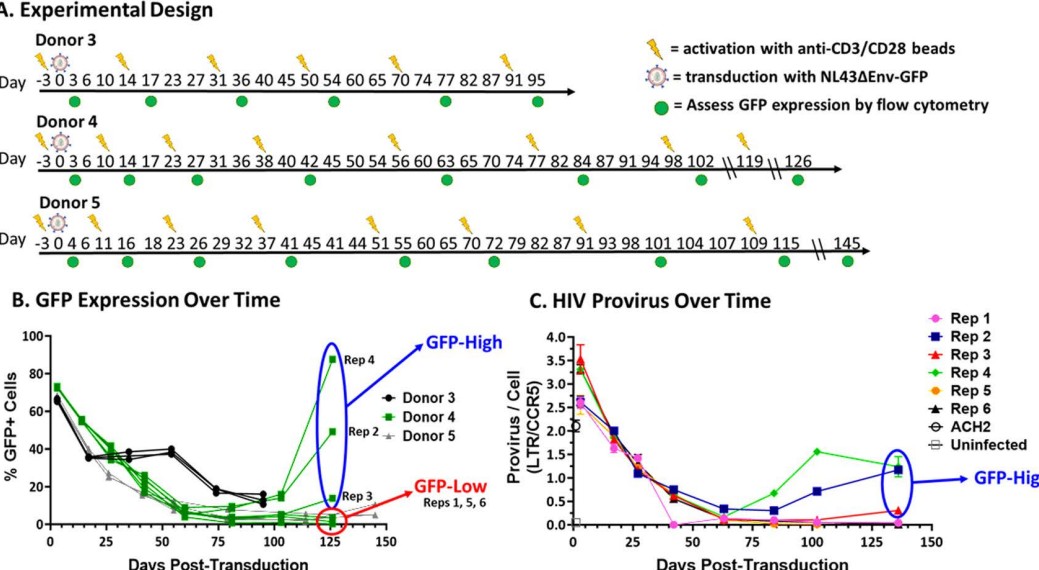

**Fig 1. Proviral integration associated with HIV+ cell survival.** A) Experimental design. Primary human CD4 + CD45RA + T cells from three different donors were infected with NL43ΔEnvΔNef-GFP pseudotyped with VSV-G at an MOI of 1 GFP TU/cell. Cells were stimulated with anti-CD3/CD28 beads as they began to die off after each expansion period. We acknowledge that BioRender was used for depiction of an HIV virion. Created in BioRender. Murimi-worstell, D. (2025) https://BioRender.com/wgmt54w. B) GFP expression of each replicate from the 3 donors over time assessed by flow cytometry. C) Bulk HIV provirus content was assessed by LTR-specific qPCR at the time points shown using ACH-2 cells as a single-copy HIV standard and CCR5 as a cellular DNA control. The provirus presence was assessed until day 136.

Our analysis showed a reduction in HIV provirus content over time in all Donor 4 replicates, reflecting a loss of infected cells rather than of provirus silencing (Fig 1C). Additionally, the subsequent increases in Gfp expression in replicates 2, 3 and 4 were accompanied by increases in provirus per cell, reflecting preferential growth or survival of an initially small fraction of infected cells (Fig 1C).

These data suggest a growth or survival advantage of an initially small fraction of the HIV infected and Gfp expressing cells, as the HIV/GFP+ cells made up increasingly large proportions of the cells that survived until day 126. To distinguish the three Donor 4 replicates with resurgent HIV/GFP+ populations, they will be referred to as "GFP-High" and the other three replicates without a visible resurgent population will be referred to as "GFP-Low."

## Resurgent HIV+ cells contain clonal proviruses within intron 1 of *STAT3*

As our laboratory has previously described [43], HIV integration within *STAT3* can be reproducibly associated with clonal expansion of primary human CD4 + T cells infected and grown in vitro. We therefore hypothesized that the HIV/GFP+ cells in the GFP-High replicates would also contain proviruses integrated in *STAT3*. To test this hypothesis, DNA was extracted from the Donor 4 day 126 samples and ISA was performed using a modification of the protocol of Wells *et. al.* [44].

The day 126 DNA samples were sheared using an enzymatic reaction to create random, unique, double strand breaks in the genomic DNA, as previously described [55]. Following a method like that previously described [44,55], a linker was ligated to the DNA fragments, which were amplified and barcoded for Illumina sequencing. Sequence data were analyzed using an analytical pipeline that trims each read of linker and proviral sequences leaving only the host sequence, which was then aligned to the hg19 reference. Random shearing of the host sequences leaves each with a unique terminal breakpoint, which will almost always be different for each starting DNA molecule. Sequences with identical integration

sites and different breakpoints must have come from different descendants of a single infected cell, which subsequently underwent clonal expansion. The number of different breakpoints for a given integration site is a measure of the relative size of each such clone.

ISA of all the Donor 4 replicates at day 126 identified integration sites throughout the genome (Fig 2A). All replicates that exhibited clonal GFP+ expansion contained proviral Integrations within STAT3. In individual replicates, proviruses within CKAP2L, DCUN1D4, GSDMD, and CCDC77 were also associated with clonal expansion at day 126.

Proviral integration patterns and GFP percentages were compared to day 17. As shown in Fig 1, there was initial decrease in GFP+ cells over time indicating selection against HIV/GFP+ cells under our culturing condition, until a resurgence in GFP-High replicates. We also observed a reduction in proviral integrations within genic regions over time, except in the GFP-High replicates where integration within specific genes made up a large portion of integration sites (clonal2B). Excluding the GFP-High replicates, we saw selection for integration within genic regions fall from about 60% to closer to 20%, corresponding to the reduction in GFP-expressing cells, which declined from around 50% to 1% (Fig 2B). Increased integrations in genic regions at initial infection followed by selection against those genic integrations over time supports findings from prior studies [38]. In the GFP-High replicates, the increase in GFP-percentage corresponded to increases in proviral integration within specific genes (Fig 2B), providing evidence that HIV integration within specific genes was associated with a survival advantage.

The GFP-High populations in replicates 2 and 4 contained clonal proviruses within STAT3, which comprised 24.3% and 8.9% of the total breakpoints, respectively (Fig 2B). The proviruses within STAT3 were all in the same orientation of transcription as the gene. Each of these two replicates also contained additional clonal integration sites (Table 1 and Fig 2A and 2B). Replicate 2 contained a clonally amplified provirus within CKAP2L, but in the opposite transcriptional orientation. Replicate 4 contained two additional genes with clonally amplified integration sites, DCUN1D4 and GSDMD, which made up 23.2% and 26.3% of total integration sites, respectively (Table 1 and Fig 2B). The amplified provirus within DCUN1D4 was in the same orientation as translation but the one in GSDMD was not (Table 1). However, the GSDMD provirus was in the same orientation as and just upstream of an annotated anti-sense lncRNA of unknown function, LOC10030756. Cells containing these additional proviruses were amplified to a similar extent as one in STAT3 and could also be contributing to the resurgence of HIV+ cells, although their possible joint contribution has not been determined.

Additional clones of cells with proviral integration sites within STAT3 were also identified in replicates 1 and 3, both of which contained fewer HIV+ cells at day 126 than replicates 2 and 4. Both replicates 1 and 3 contained proviruses within STAT3, which made up just under 2% of total integrations in each (Table 1 and Fig 2B). Additionally, however, replicate 3 contained 18.4% of total integrations within CCDC77, suggesting that clonal expansion of cells with a provirus at this site may also have played a role in the 13% GFP+ cells (Table 1 and Fig 2B).

STAT3 integration sites identified in the four replicates were in the same orientation of transcription as the STAT3 host and most were in intron 1 just before the start of exon 2, the first coding exon (Fig 2C). HIV provirus integration within this region of STAT3 has previously been described in cases of HIV-associated T cell and B cell lymphomas [16,17]. Replicate 3 contained integration sites within introns 1 and 3 (Table 1 and Fig 2C), the latter being the location of some of the previously described integrations influencing clonal expansion of T cells grown in vitro and potentially leading to a shortened protein product via N-terminal truncation [43].

PCR was performed on all replicates to confirm the presence of HIV proviral integration sites identified by ISA. Primers complementary to different locations within intron 1 of STAT3 and the HIV 3'LTR identified integration sites within the GFP-High replicates 2, 3 and 4 (Fig 2D). Together with the ISA data, proviruses integrated in intron 1 of STAT3 were associated with resurgent GFP/HIV+ populations within these three Donor 4 replicates. Proviral integration within STAT3 was not detected in replicate 1, most likely because there were only 3.5% HIV/GFP+ cells at time of DNA collection. Detection of proviruses in STAT3 associating with clonal expansion implies that HIV integration within the first intron of STAT3 can contribute to the survival, outgrowth, and persistence of HIV+ cells in the culturing conditions shown in Fig 1A.

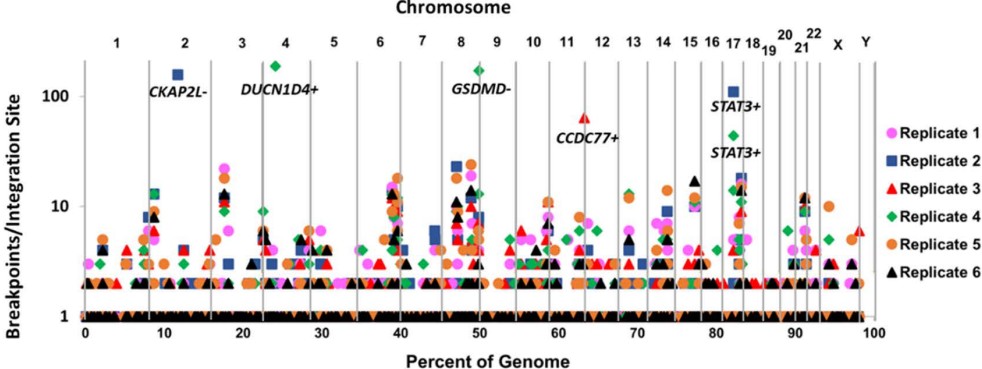

## A. Integration Site Analysis of Donor 4 Replicates

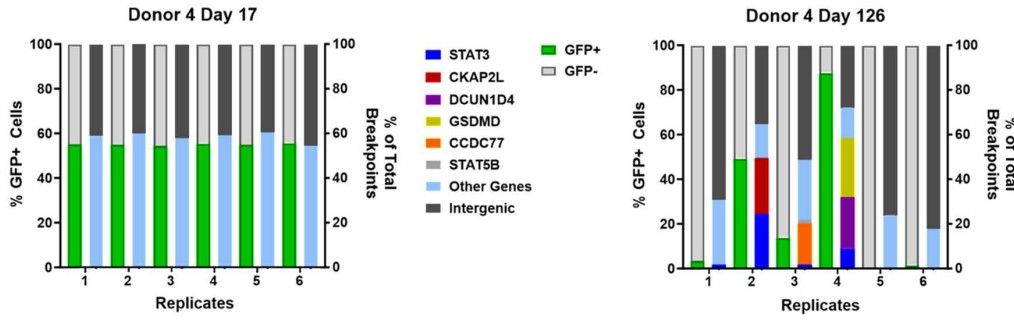

## B. Characterization of Proviral Integration Sites

## C. Proviral Integrations within *STAT3*

## D. Confirmation of *STAT3* integration Sites

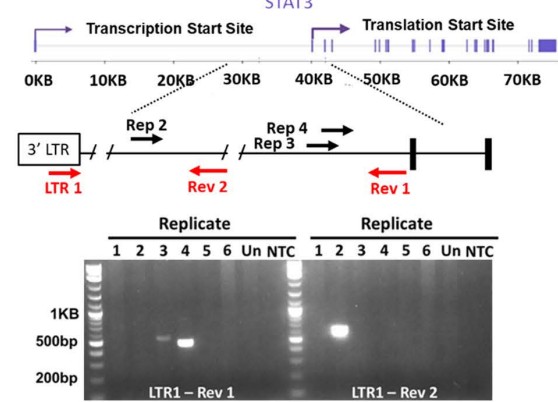

**Fig 2. Integration in *STAT3* is associated with GFP-High replicates.** A) Integration site analysis was performed on all six Day 126 Donor 4 replicates and is shown as the number of different breakpoints for each site (Y axis) vs the chromosomal location of the integration site within the whole human genome (X-axis). Integration sites with large numbers of breakpoints that contained additional sites in immediate proximity to one another were combined for a true representation of clonality [56] and are indicated with the relative orientation of the provirus at their chromosomal location. B) Relative frequency of integration sites in the indicated genes in the different replicates at Day 17 and Day 126. Left Y-axis measures percentage of GFP positive cells. Green represents GFP+ cells and light grey represents GFP- cells. The right Y-axis measures percentage of total breakpoints. Percentage of integrations within intergenic regions are in dark grey. Integrations within genes are colored light blue, and integrations within genes associated with clonal expansion are indicated by the color shown in the key for each individual gene. C) Location, orientation, and frequency of proviral integration sites within *STAT3* of the replicates. D) Top: structure of *STAT3* and location of primers (red arrows) used to confirm the integration sites shown by the black arrows within intron 1. Bottom: gel analysis of products of PCR using the STAT3 and LTR primers to confirm integration sites identified in replicates 2, 3, and 4 by ISA. Un, uninfected T cells; NTC, no template control.

**Table 1. Clonal proviral integration sites in donor 4 replicates at day 126 after infection.**

| Replicate | % GFP | Gene | Orientation to Gene | Integration Sites (hg19) | Breakpoints | % of Total Breakpoints |
|---|---|---|---|---|---|---|
| 1 | 3.50% | *STAT3*, Intron 1 | + | chr17, 40500758 | 5 | 1.7% |
| | | Other genes | N/A | N/A | 87 | 29.0% |
| | | Intergenic | N/A | N/A | 208 | 69.3% |
| 2 | 49.2% | *STAT3*, Intron 1 | + | chr17, 40506456 | 109 | 24.3% |
| | | *CKAP2L*, Intron 3 | − | chr2, 113515517 | 113 | 25.2% |
| | | Other Genes | N/A | N/A | 68 | 15.1% |
| | | Intergenic | N/A | N/A | 159 | 35.4% |
| 3 | 13.8% | *CCDC77*, Intron 7 | + | chr12, 540186 | 44 | 18.3% |
| | | *STAT3*, Intron 1 | + | chr17, 40501333 | 4 | 1.7% |
| | | STAT3, Intron 3 | + | chr17, 40497665 | 1 | <0.1% |
| | | STAT5B, Intron 1 | + | chr17, 40426185 | 4 | 1.7% |
| | | Other Genes | N/A | N/A | 64 | 26.8% |
| | | Intergenic | N/A | N/A | 123 | 51.3% |
| 4 | 87.6% | *DCUN1D4*, Intron 1 | + | chr4, 52727751 | 146 | 23.2% |
| | | *GSDMD*, Intron 2 | − | chr8, 144641746 | 166 | 26.3% |
| | | *STAT3*, Intron 1 | + | chr17, 40501270 | 42 | 6.7% |
| | | *STAT3*, Intron 1 | + | chr17, 40505628 | 14 | 2.2% |
| | | Other Genes | N/A | N/A | 86 | 13.6% |
| | | Intergenic | N/A | N/A | 176 | 27.9% |
| 5 | 0.30% | Other Genes | N/A | N/A | 76 | 23.8% |
| | | Intergenic | N/A | N/A | 243 | 76.2% |
| 6 | 1.40% | Other Genes | N/A | N/A | 31 | 17.7% |
| | | Intergenic | N/A | N/A | 144 | 82.3% |

## Enrichment of STAT3 expression in Donor 4 replicates with increased HIV/GFP expression

To understand the influence of proviral integration on expression of *STAT3* and its downstream target genes, bulk RNAseq was performed at day 126 on all GFP-High and GFP-Low replicates from Donor 4. The GFP-High replicates, which had greater than 10% GFP+ cells on day 126, expressed higher levels of *STAT3* than the GFP-Low replicates, with less than 10% of cells expressing *gfp* (Fig 3A). Replicate 2 showed the highest expression of *STAT3*, and the highest percentage of integration sites within *STAT3*, but did not have the highest percentage of GFP+ cells. This result hints that integration within genes other than *STAT3*, or possibly other epigenetic effects, may also contribute to HIV/GFP+T cell survival and persistence.

Previous studies looking at HIV-1 proviral integrations sites in vivo, have identified that the HIV LTR can act as a promoter and drive the expression of host genes [37,45,46]. To determine if *STAT3* overexpression was driven by the HIV LTR, we compared the number of reads from exons 2–24, downstream of the integration site, to those from exon 1, upstream of the integration site (Fig 3B). The GFP-High replicates showed an increase in the ratio of exon 2 reads to exon 1 reads compared to the GFP-Low replicates (S2 Fig). This increase provides evidence that *STAT3* overexpression was driven by the HIV LTR. Taken together, these results imply that LTR driven chimeric transcripts of *STAT3* were driving *STAT3* overexpression in the GFP-High replicates, thus providing additional evidence that insertional mutagenesis of *STAT3* led to the growth advantage of the HIV/GFP+ cells.

To explore the effects of *STAT3* expression in resurgent GFP+ cells, differential gene expression analysis was performed using DESeq2 [58]. For this analysis, the Donor 4 replicates were divided into GFP-High (Reps 2, 3, and 4) and GFP-Low (Reps 1, 5, and 6) groups, and data for each group were pooled for analysis of differentially expressed genes

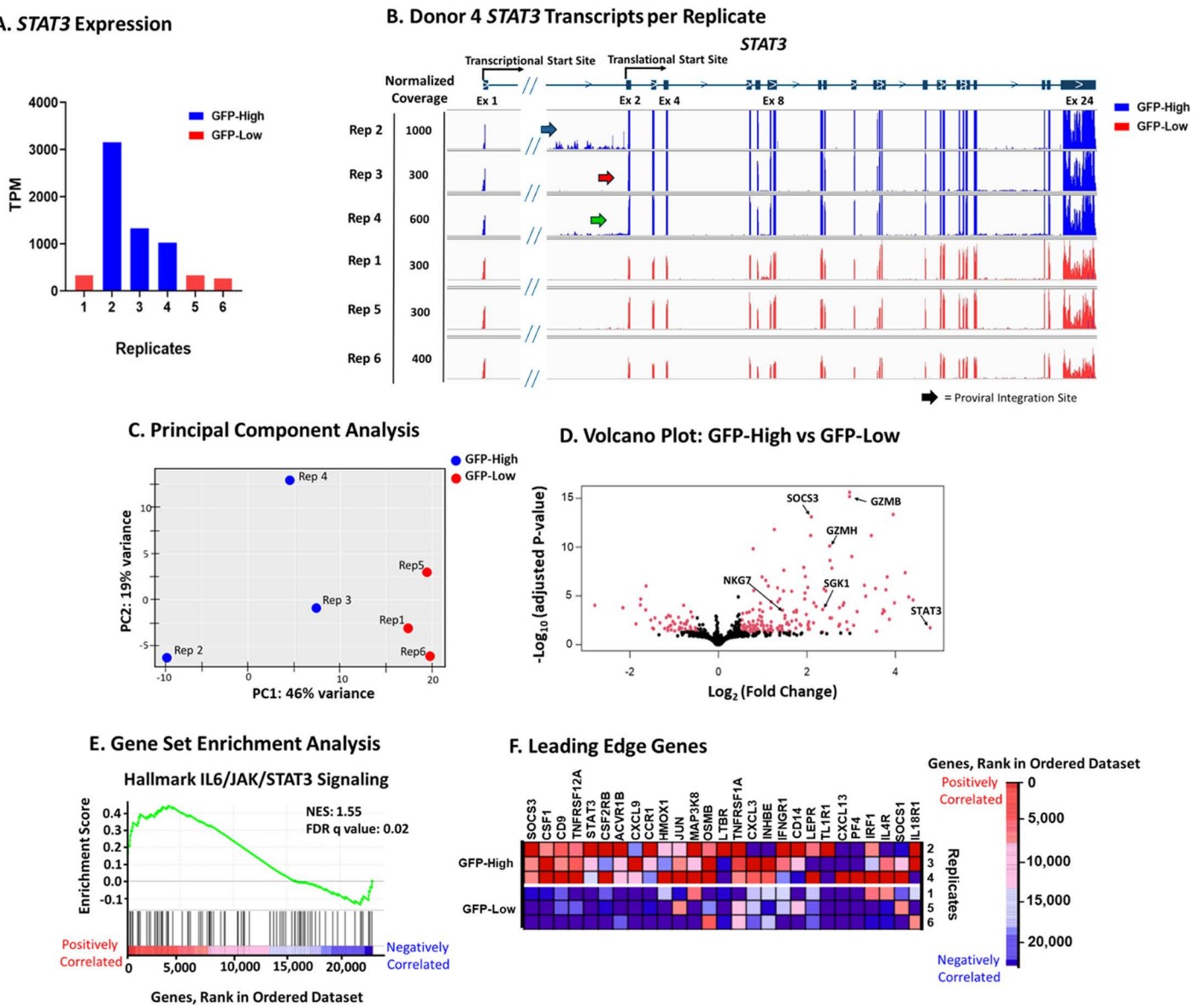

**Fig 3. Upregulated *STAT3* signature in GFP-High replicates.** A. Bulk RNA-seq of unsorted donor 4 replicates was performed at day 126 post-transduction. Replicates 2,3, and 4 were designated as "GFP-High" and Reps 1,5, and 6 as "GFP-Low." A) *STAT3* expression in GFP-high (blue) and GFP-low (red) Donor 4 replicates in transcripts per million (TPM). B) Coverage of transcripts across *STAT3* in each replicate of Donor 4. C) Principal component analysis using DESeq2 (56) of GFP-High replicates compared to GFP-Low replicates. D) Volcano plot of GFP-High gene expression compared to GFP-Low gene expression. Genes that are differentially expressed are labeled in red. Parameters for differential expression are log2 fold change > 0.5 and adjusted p-value < 0.05. E) IL6-JAK-STAT3 Signaling hallmark gene set, from the Molecular Signatures Database (MSigDB), analyzed by gene set enrichment analysis [57] normalized enrichment score (NES) = 1.55, FDR q-value = 0.02. F) Heat map of gene rank of the leading-edge genes in the Hallmark IL6/JAK/STAT3 signaling gene set.

(Fig 3C). Overall, there were only 213 genes differentially expressed (adjusted p < 0.05) between the two groups (Fig 3D). Notably, *STAT3* was one of the most differentially expressed upregulated genes (Fig 3D). Additionally, downstream targets of STAT3, such as *SOCS3* [59,60] and *SGK1* [61], were also upregulated (Fig 3D). All three of these STAT3-related genes are associated with anti-apoptotic pathways [62–65].

GZMB, encoding granzyme B, a serine protease associated with cytotoxic T cells, was identified as one of the most highly differentially expressed genes in replicates 2 and 3 (S3 Fig), in addition to GZMH and NKG7, two other markers of cytotoxic T cells (Fig 3D). Previous studies have identified STAT3 as a transcription factor for GZMB [66–68], suggesting that insertional mutagenesis of STAT3 in our system led to an increase in its expression. The upregulation of these markers of cytotoxic T cells is of particular interest, since their expression has been identified as being associated with persistent HIV-infected CD4 + cells from PWH, which were enriched in expression of a Th1 cytotoxic cell phenotype [41,42].

To further compare GFP-High replicates to the GFP-Low replicates, gene set enrichment analysis (GSEA) [57] was performed. GSEA looks for enrichment in predetermined sets of genes between the groups being compared. Utilizing the 50-hallmark gene sets from the molecular signatures database (MSigDB), GFP-High replicates were significantly enriched in the hallmark gene set IL-6/JAK/STAT3 signaling (Fig 3E and 3F). Upregulation of STAT3 and STAT3-related genes and enrichment of the STAT3 signaling pathway (NES 1.55, q value 0.02) provides evidence that the LTR-driven STAT3 plays an active role in shaping cellular gene expression leading to preferential growth and/or survival of the affected cells from the Donor 4 GFP-High replicates.

## GFP-High replicate 2 has largest increase in protein expression and phosphorylation of STAT3

To confirm that upregulation of the STAT3 transcription signature corresponds to upregulation at the protein level, cells frozen at earlier time points were thawed and cultured until day 126 and analyzed by flow cytometry for Gfp and STAT3 expression and activation. GFP-High replicates contained distinct populations of GFP+ cells with higher expression of STAT3 compared to the GFP- cells, whereas the GFP-Low sorted replicates did not contain these overexpressing STAT3 populations (Fig 4A). Replicate 2 showed the largest populations of STAT3 overexpressing cells with Replicates 3 and 4 having smaller populations. The increased protein expression reflects the increase in STAT3 transcription between the GFP-High replicates (Figs 3A and S2). Moreover, the staining for STAT3 identifies the number of cells overexpressing STAT3, by increased fluorescence intensity, which quantifies the number of HIV/GFP+ cells where insertional mutagenesis of STAT3 is driving survival. Thus, replicate 2 had the largest percentage of integration within STAT3 (Table 1 and Fig 2A and 2B), the highest expression of STAT3 transcripts (Fig 3A and 3B), and the largest population of overexpressing STAT3 protein (Fig 4A and 4B). Likewise, insertional mutagenesis of STAT3 played a smaller role in the resurgent HIV/GFP+ cells in Replicate 3 and 4.

To see if the increased STAT3 was active in the GFP-High replicates, they were stained with antibody specific for phosphorylation at tyrosine 705 (Y705) of STAT3 (pSTAT3), a modification that allows STAT3 to dimerize, enter the nucleus, and carry out its role as a transcription factor. The replicates were analyzed for pSTAT3 (Y705) by flow cytometry and showed an increase in the percentage of pSTAT3-positive populations in the GFP+ cells compared to the GFP- cells (Fig 4C). However, only the GFP+ cells in replicates 2 and 3 had increased pSTAT3 expression, measured by fluorescence intensity, compared to the GFP- cells (Fig 4D), while a difference in pSTAT3 expression between GFP+ and GFP- cells was not apparent in replicate 4. Replicate 4 had a smaller percentage of HIV/GFP+ cells over expressing STAT3 and there was no difference between the GFP+ and GFP- cells, indicating some difference, perhaps integration within a different gene, in the mechanism leading to clonal expansion and survival in this replicate.

The flow cytometry data identified STAT3 overexpressing populations within the HIV/GFP+ fraction of the three GFP-High replicates. Further interrogation identified increases in phosphorylation of STAT3 within the GFP+ cells of replicates 2 and 3. Increased abundance of STAT3 might allow for the increase in activated STAT3 and therefore drive upregulation of genes that use STAT3 as a transcription factor. We conclude that overexpression and activation of STAT3 in the GFP-High replicates led to enrichment in STAT3 signaling and upregulation of downstream targets of STAT3, thereby contributing to the resurgence and survival of the HIV/GFP+ cells in the GFP-High replicates.

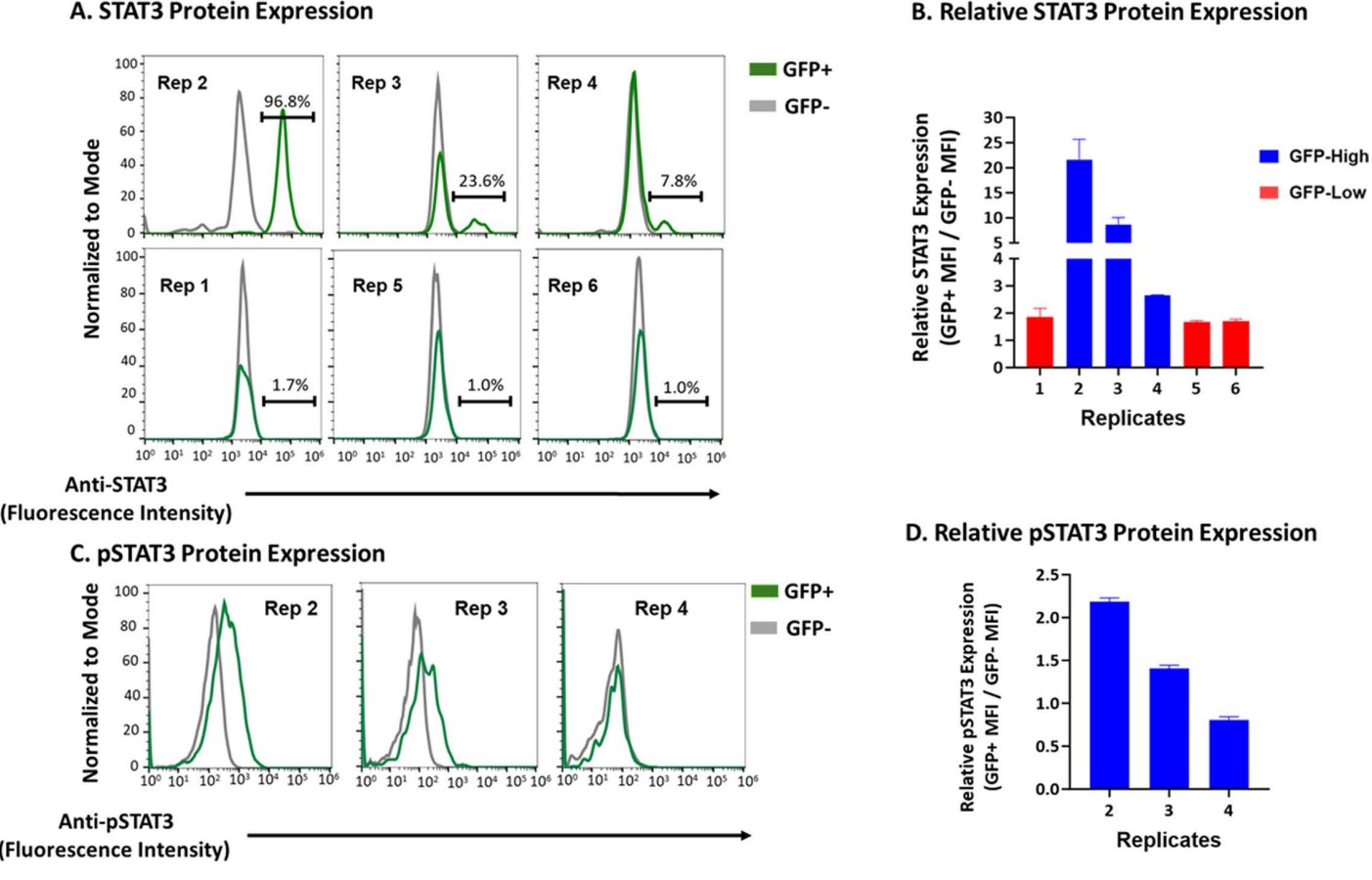

**Fig 4. Increased *STAT3* expression and activation in GFP-High Replicates.** A2) Donor 4 replicates on day 126 were stained for STAT3 expression and cell viability (using Zombie Violet™). The day 126 cells were analyzed by flow cytometry for cell viability and viable cells were then sorted for Gfp expression. STAT3 expression was then assessed in the sorted GFP+ and GFP- cells. B) Median fluorescence Intensity (MFI) was compared between GFP+ and GFP- cells from respective replicates. C) Day 126 Donor 4 GFP-High replicate cells were stained for pSTAT3 (Y705) expression and cell viability. The day 126 cells were gated on cell viability then analyzed by flow cytometry. Viable cells were then gated for Gfp expression and pSTAT3 expression was assessed in the GFP+ and GFP- cells. D) MFI compared between GFP+ and GFP- cells from respective replicates.

## STAT3 signaling is associated with persistence of replicate 2 and 3 HIV/GFP+ cells

To further investigate the increase of GFP+ cells in these populations at later time points, bulk RNAseq was performed on sorted GFP+ and GFP- cells from each Donor 4 GFP-High replicate and differential gene expression analysis was performed using DEseq2. Pooled GFP+ data were compared to the GFP- cells sorted from the GFP-High replicates (Fig 5A) resulting in 273 differentially expressed genes (adjusted p < 0.05) (Fig 5B). Of these differentially expressed genes, again, *STAT3* overexpression due to HIV proviral insertional mutagenesis was a driving factor in the HIV/GFP+ cell survival advantage.

To better understand what the differentially expressed genes reveal about the GFP+ cells, Gene Ontology (S4A Fig) and Kyoto Encyclopedia of Genes and Genomes (KEGG) Pathway (S4B Fig) analyses were performed. Both analyses identified upregulation of genes associated with DNA replication, cell division and cell cycle. These results suggest that the HIV/GFP+ persistent cells were undergoing a higher level of cell division than the GFP- negative cells. This conclusion was further supported by GSEA, which also showed significant enrichment in cell cycle associated gene sets, E2F Targets and G2M Checkpoint (S4C Fig), consistent with greater HIV/GFP+ cell proliferation and expansion.

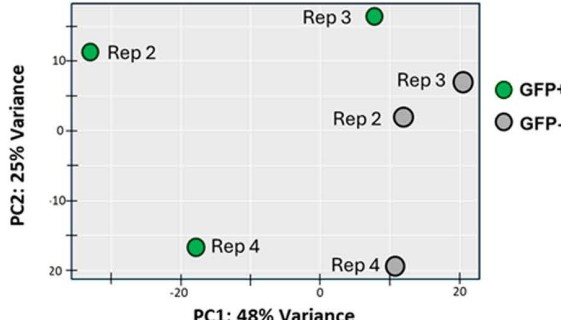

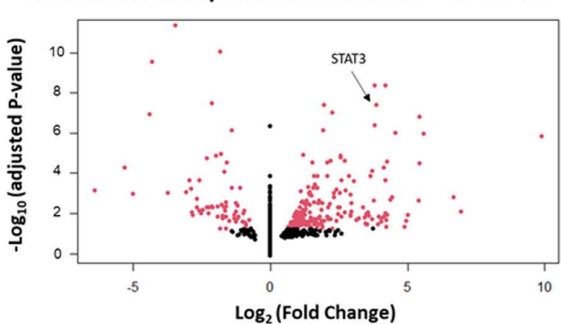

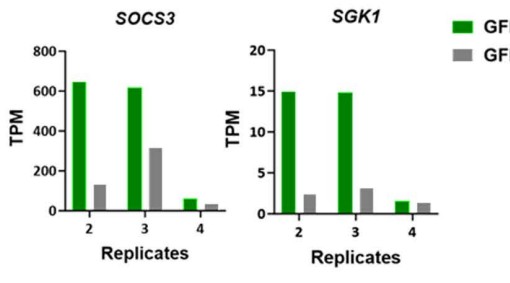

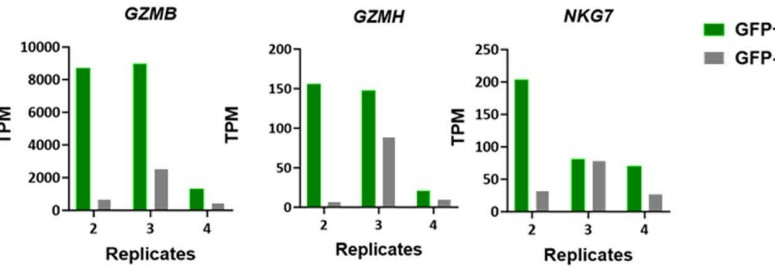

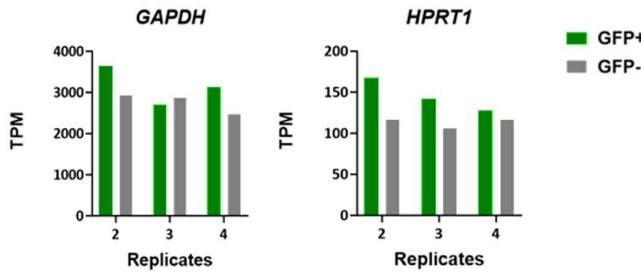

**Fig 5. STAT3 signaling plays the largest role in Replicate 2 and 3 HIV/GFP+ cell survival.** Donor 4 GFP-High replicates on Day 126 were sorted for GFP expression and viability and the two fractions were subjected to RNAseq. A) Principal component analysis using DESeq2 of sorted GFP-High cells. B) Volcano plot of sorted GFP-High replicates 2, 3, and 4. Data from GFP+ cells were pooled and compared to that from pooled XGFP- cells. Pooled data from the 3 replicates are shown. C) Expression of apoptosis-associated genes compared between GFP+ and GFP- cells from GFP-High replicates. D) Comparison of expression of cytotoxic genes. E) Comparison of expression of some housekeeping genes.

Despite *STAT3* being a differentially expressed gene, its previously identified downstream targets were not identified as such in the pooled data from the three GFP+ cells from the GFP-High replicates. Reduced STAT3 overexpression and activation in replicate 4 GFP+ cells compared to the other GFP-High replicate GFP+ cells (Fig 4A and 4C), might explain why the downstream targets of STAT3 were not identified as genes differentially expressed between GFP+ and GFP- cells. Examination of the differentially expressed genes of interest identified in Fig 3D shows that that targets of STAT3 were upregulated in replicates 2 and 3 GFP+ cells but to a lesser extent or not at all in replicate 4. *SOCS3* and *SGK1* were both increased in expression only in the two replicates with increased STAT3 expression and activation (Fig

5C). Further suggesting that increases in these anti-apoptotic genes were contributing to T cell persistence. Additionally, STAT3-related cytotoxic factors, *GZMB* and *GZMB*-related genes, were also upregulated in GFP+ cells from replicates 2 and 3 but not replicate 4 (Fig 5D). GZMB was upregulated in GFP+ cells when analyzed by flow cytometry (S3 Fig). All three of the GFP-High GFP+ replicates contained a higher percentage of GZMB+ cells and a higher mean fluorescence intensity than their matching GFP- cells. GFP+ cells in replicate 2 showed the largest increase in GZMB compared to GFP- cells (S3 Fig). Control genes showed no difference (Fig 5E).

Identification of *STAT3* as a shared differentially expressed gene further supports the role of STAT3 in cell survival and/or proliferation. STAT3 activity is most commonly associated with IL-6 or IL-10 signaling [69] but also plays roles in other pathways such as response to TCR engagement [70]. The primary CD4 + T cell cultures were stimulated by TCR signaling through anti-CD3/CD28 beads, as conclusion supported by enrichment in cell cycle and proliferation in gene ontology, pathway, and gene set enrichment analyses (S4 Fig). However, only replicates 2 and 3, which had increased STAT3 activation, showed a clear activated STAT3 expression signature (Fig 4C and 4D). Taken together, the data suggest that upregulation of activated STAT3 can influence anti-apoptotic gene signatures, which might contribute to primary T cell persistence in culture.

Considering the role of STAT3 as a transcription factor in T cell development, we utilized RNAseq data to analyze subtypes of the T cells. The patterns of gene expression of markers of T cell subtypes implied that the T cells grown in culture did not seem to be influenced by *STAT3* expression (S5A Fig). In general, *STAT3* expression is most associated with the Th17 subtype [70]. However, gene signatures more associated with Th2 subtypes were associated with both the GFP/HIV+ and GFP- groups. This T cell subtype signature has been recorded with T cells that are activated in culture with TCR stimulation but do not receive a secondary signal, such as IL6 for Th17 [71]. This result is consistent with the use, in our model, of CD3 and CD28 crosslinking to mimic TCR stimulation without a secondary signal. Therefore, our persistent cells at day 126 expressed a Th2-like signature instead of the Th17 signature associated with *STAT3* expression in vivo.

STAT3 has also been implicated in T cell phenotype development from naïve to memory [27,60]. RNA sequencing profiles of the GFP/HIV + GFP-High replicates did not show many stark changes to a memory or naïve phenotype compared to the GFP- cells (S5B Fig). Flow cytometric analysis of our GFP-High replicates was performed by staining for CD45RO, a marker of memory T cells [72,73], as well as CCR7 which is expressed on central memory ($T_{CM}$) and absent on effector memory ($T_{EM}$) T cells [72,73]. The results revealed that the GFP+ cells in replicates 2 and 3, which had larger clonal populations with *STAT3* integrations, had a greater increase in both $T_{CM}$ and $T_{EM}$ differentiation than the GFP- cells from the same replicate (S5C Fig). Replicate 4 GFP/HIV+ cells only demonstrated increase in $T_{EM}$ (S5C Fig). Taken together, STAT3 may be driving differentiation of primary T cells into a memory phenotype, but as these T cells are grown in vitro with no secondary signals further exploration would be needed to support these findings.

## Resurgent HIV+ donor 4 cells do not contain unique STAT3 polymorphisms

We saw an increased STAT3 expression in three replicates of donor 4 associated with increased GFP + T cells but we did not see this effect in cells from the other two donors transduced in vitro. Therefore, we asked whether Donor 4 had any specific polymorphisms that may have influenced *STAT3* activity. Utilizing RNAseq data from the Donor 4 replicates, *STAT3* variants were identified by comparison to the hg38 genome (S1 Table). Of note, there were two T to C polymorphisms in all replicates located in exon 24, which is an untranslated region of *STAT3*. While more polymorphisms were detected in expressed introns in replicate 2, all the variants were also found in other replicates, therefore there were no polymorphisms that were specific to *STAT3* in resurgent replicates influencing cell growth or survival and there were no modifications to the active sites, SH2 domain or DNA binding region, of STAT3. However, polymorphisms in the 3'UTR can alter the function of *STAT3* transcripts through effects on post-transcriptional processing. While these changes are not obvious alterations in coding exons they could lead to alterations in processes, such as mRNA stability, of STAT3 that contributed to survival of T cell clones with proviruses in *STAT3*.

## Integration in genes other than STAT3 may contribute to HIV/GFP+ cells survival advantage

Flow cytometry, shown in Fig 4A, identified STAT3 overexpression in only a portion GFP+ cells in replicates 3 and 4. To investigate the potential of proviruses at other integration sites to contribute to HIV/GFP+ survival, day 126 cells were sorted for Gfp expression and cell viability. DNA and RNA were extracted from both the GFP+ and GFP- populations of each GFP-High replicate for analysis by qPCR and bulk RNA sequencing. qPCR was performed to assess the number of proviruses present in both the GFP+ and GFP- cells. Quantification of HIV proviruses allows us to understand if the effects of proviral integration work synergistically with multiple integrations per cell or separately with cells containing single proviruses at different locations. Using the same qPCR approach as Fig 1C, GFP+ cells in replicates 2 and 3 were found to contain around 2 HIV proviruses per cell, and those in replicate 2 contained closer to 3 proviruses per cell (Fig 6A). These results imply that, in replicates 3 and 4, two proviruses were influencing individual cell survival. On the other hand, replicate 2 HIV/GFP+ cells contained an additional provirus per cell and multiple proviruses in *STAT3* (Fig 2C), suggesting that a large fraction of these cells contained proviruses integrated at additional sites along with one in *STAT3*. FACS analysis showed that most replicate 2 cells were overexpressing STAT3, implicating it as the main contributor to HIV/GFP+ cell persistence (Fig 4A).

These results implied that the clonally expanded cells with a provirus in *STAT3* also had a second provirus in another gene, possibly contributing to their expansion. To understand the role of integration sites associated with clonal expansion in HIV/GFP+ cell persistence, we used bulk RNAseq analysis on the FACS sorted GFP+ and GFP- populations for each GFP-High replicate to investigate transcriptional changes in genes with proviruses showing large numbers of breakpoints. Increased expression of STAT3 was identified in the GFP+ relative to GFP- cell population in all three GFP-High replicates (Fig 6B). Clonal integration in the same orientation as gene transcription as well as increased expression of *CCDC77* was observed only in GFP+ replicate 3 cells, while increased integration in and increased expression of *DCUN1D4* were seen in in GFP+ replicate 4 cells (Fig 6B). Increases in these transcripts were observed just downstream the location of proviral integration, providing evidence that the increased expression was driven by insertional mutagenesis (Fig 6C). Unfortunately, little is known about the roles of *CCDC77* and *DCUN1D4* in cell function, making it difficult to infer how HIV insertional activation of expression of these genes might be affecting cell persistence.

In the case of replicate 2, GFP+ cells contained clonal HIV provirus integration in *CKAP2L* (Fig 2A and 2B and Table 1). However, no difference was seen in CKAP2L expression between replicate 2 and the other GFP-High replicates (Fig 6D). Examination of expression in IGV also revealed significant increases in levels of *CKAPL* transcripts downstream of the site of HIV integration same orientation as the provirus but, in this case, in opposite orientation to the gene (Fig 6E, blue). Altogether, these data suggest that HIV/GFP+ cells in replicate 2 contain multiple proviruses influencing host RNA expression. Therefore, the effects of multiple proviruses integrated in the HIV/GFP+ cells in this replicate could contribute to their survival through unique mechanisms.

Another expanded GFP+ clone also contained a provirus antisense to the host gene, in this case, in the second intron of *GSDMD*, encoding gasdermin D, in replicate 4 (Fig 6E). This provirus was also upstream of and in the same sense as *LOC100310756,* a lncRNA of unknown function (Fig 6E). As with the other proviruses in the clonally expanded GFP+ cells, the *GSDMD* provirus was associated with greatly increased expression of *LOC100310756*, but, again, transcription of the overlapping *GSDMD* was not significantly affected

Several studies have found that early activation of the HIV protease by some non-nucleoside reverse transcriptase inhibitors, including efavirenz (EFV), can induce GSDMD mediated pyroptotic cell death in T cells and macrophages by premature activation of HIV protease leading to cleavage of CARD8 and driving caspase 1 activation and inflammasome production [74–76]. To see if replicate 4 clones, showed resistance to this cell death pathway due to integrations within *GSDMD*, we cultured early (Day 31) and late (Day 114) time points of replicate 4 cells with DMSO, EFV, or lopinavir (LPV). LPV is an HIV protease inhibitor, which has been shown to prevent this pyroptotic cell death pathway [74,76]. After 72 hrs of culturing cells with respective anti-retrovirals, late time point replicate 4 had little reduction, around 10%, in GFP/HIV+ cells, whereas the

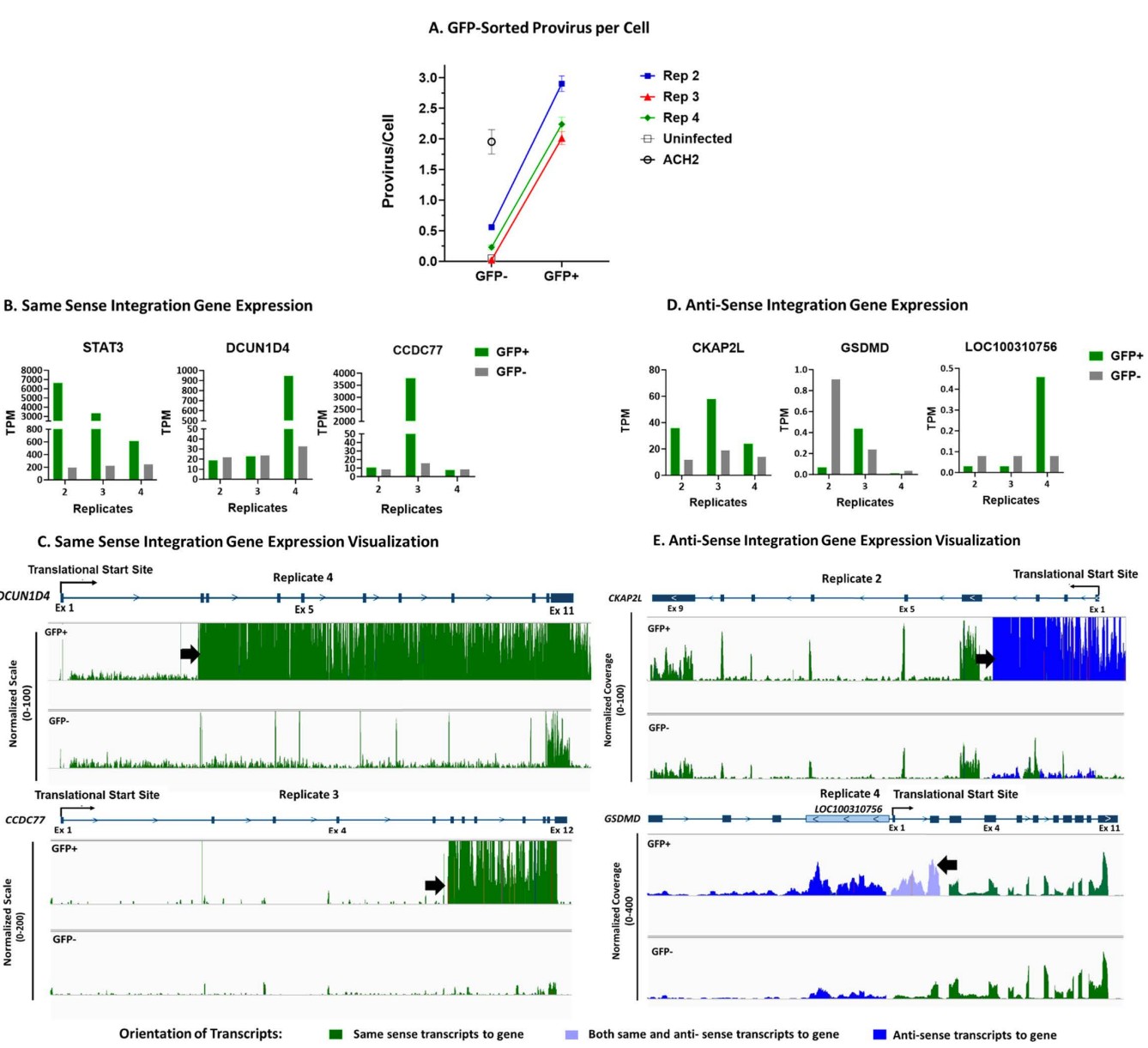

**Fig 6. Increased *STAT3* expression and activation in GFP-High Replicates.** DNA and RNA were extracted from GFP-High replicates sorted for cell viability and GFP expression on day 126. A) qPCR was performed to estimate provirus copy number per cell using primers and probes for LTR and normalized to host CCR5 sequences. ACH-2 cells were used as standard and day 126 uninfected cells as negative control. B-E) Bulk RNA-seq counts were used to assess transcriptional activation related to the provirus insertion. Sorted GFP+ samples are colored green and GFP- samples are colored in grey. B) Expression of genes in the indicated replicates with same sense proviruses associated with clonal expansion. Gene expression from the RNAseq analysis was compared between GFP+ and GFP- samples. C) Sequence coverage of *STAT3* as well the genes *DCUN1D4* and *CCDC77*, which have a provirus integrated in the same transcriptional sense as the host gene in replicates 4 and 3, respectively. The peak heights are all normalized relative to exon 1. Translational start sites are identified on the gene maps and integration sites are shown by black arrows. D) Expression of genes with provirus integration in opposite orientation to a host gene associated with clonal expansion. Integration in replicate 4 is in same orientation as *LOC100310756* but between exons 2 and 3 and in opposite orientation within *GSDMD*. E) Top panels. Sequencing coverage of *CKAP2L*, which had an opposite sense orientation integration in exon 5 of replicate 2. The translation start site is identified in the map and the integration site is identified with an arrow. Bottom Panels. Sequencing coverage of *GSDMD*, which had an opposite sense orientation integration in replicate 4. The color of the coverage graphs represents the predominant transcripts that align with the gene. Green represents transcripts that have the same sense alignment with the gene, blue are anti-sense transcripts to the overall gene, and the light purple represents transcripts of both same sense and anti-sense to overlapping genes. *LOC100310756* transcripts are colored blue as the transcript is anti-sense to *GSDMD*.

early time point cells lost about 50% of GFP/HIV+ cells (S6 Fig). The limited reduction in the late time point replicate 4 HIV+ cells, suggests that proviral integrations in *GSDMD* and/or integration upstream of *LOC100310756*, influenced T cell survival. The early time points were before the clonal population dominated the GFP/HIV+ cell populations Fig 2B and 2C), further supporting that the proviral integrations within this genic region could be influencing survival. While the level of *GSDMD* transcript did not seem to be greatly affected, translation of *GSDMD* may have been affected by the antisense RNA or by the increase in *L0C100310756*, also antisense to the *GSDMD* gene locus. These results are consistent with an effect on CARD8 inflammasome mediated cell death, but further studies would be needed to validate these findings.

## Persistent HIV+ cells share many characteristics with HIV-associated T Cell lymphomas

As previously discussed, Mellors *et. al.* [17] described several HIV+T cell lymphomas with clonal provirus integration at sites in *STAT3* intron 1, near those in the GFP-High replicates described here. Our next step was to understand how persistent HIV/GFP+ cells, especially from replicates 2, 3, and 4, compared to the HIV+T cell lymphomas. Publicly available bulk RNAseq data from the two high-grade T cell lymphomas [17], were available for comparison with the infected GFP-sorted primary T cells from Donor 4. DEseq2 was used for differential gene expression analysis. Principal component analysis showed that the lymphomas differed greatly in PC1 from both the HIV/GFP+ and GFP- sorted T cells infected and grown in vitro (Fig 7A). These transcriptional differences could be due to a variety of factors such as heterogeneity of tumor cell populations and further events, including additional mutations or HIV proviruses at other integration sites [17] in the samples compared. Nevertheless, to understand if the GFP-High replicates model these HIV+T cell lymphomas, we compared the pattern of differentially expressed genes in GFP-High replicates relative to the GFP-Low replicates compared to the lymphomas (Fig 7B). *STAT3* and its downstream targets *SOCS3* and *SGK1* were upregulated in the Replicate 2 and 3 GFP+ cells compared to their corresponding GFP- cells. Cytotoxic T cell markers such as *GZMB*, *GZMH* and *NKG7,* were also upregulated. Mellors *et. al.* [17] also noted the increased expression of *STAT3, GZMB* and *TNFRSF8* (CD30) as important markers of the HIV+ lymphomas.

Increased STAT3 activity, GZMB expression, and CD30 expression are markers of anaplastic large cell lymphomas (ALCL) of T cell origin, which PWH are of increased risk of developing [20]. Mellors *et. al.* [17] have also shown clonal proviral integrations within *STAT3* from DNA extracted from formalin-fixed paraffin-embedded samples of T cell ALCL. ALCLs are distinguished by CD30 expression and usually driven by mutations in the receptor tyrosine kinase ALK [77]. However, ALK-negative ALCLs, which are driven by *STAT3* dysregulation and express cytotoxic factors, are also the type of ALCL described with clonal HIV integrations. Using markers of ALK- ALCL we compared the GFP-sorted Donor 4 replicates with the high-grade T cell lymphomas to see if any of the Donor 4 replicates might model HIV-driven lymphoma (Fig 7C). This comparison confirmed that all samples lacked *ALK* expression. However, the upregulation of the T cell activation and ALCL marker CD30 (*TNFRSF8*) was present in all samples, including GFP+, GFP- and tumor samples. While many of the genes associated with ALCL were expressed in the Donor 4 replicates, previously mentioned genes, *TNFRSF8*, *STAT3*, *GZMB*, and *PRF1* stood out as upregulated in the resurgent replicate 2 HIV/GFP+ cells and high-grade T cell lymphoma samples. Additionally, clusterin (*CLU*) expression was also upregulated in both the GFP+ samples, particularly in replicates 1, 2 and 3, and the lymphomas. Clusterin is an anti-apoptotic cell survival marker [78] that has been characterized in oncogenesis [79]. These comparisons of the GFP-High replicates to the HIV+ high-grade T cell lymphoma samples suggest that the GFP-High replicates 2 and 3 model at least some oncogenic pathways and T cell phenotypes contributing to the transformation of the HIV+T cells in vivo.

## Discussion/Conclusion

Our study showed that HIV insertional mutagenesis of host genes, particularly *STAT3*, in primary human CD4+T cells can provide a survival advantage to infected CD4+T cells grown in culture. Utilizing integration site analysis and long-term cell culture, we found HIV proviruses integrated within the first intron of *STAT3* in the 3/6 replicate cultures that had an increase in HIV/GFP+ cells, following an initial period of loss of infected cells.

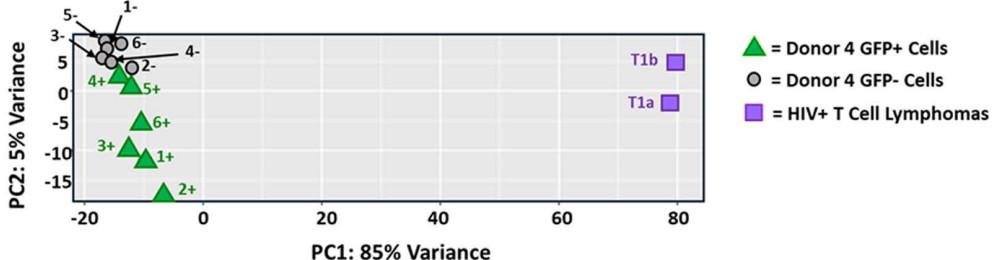

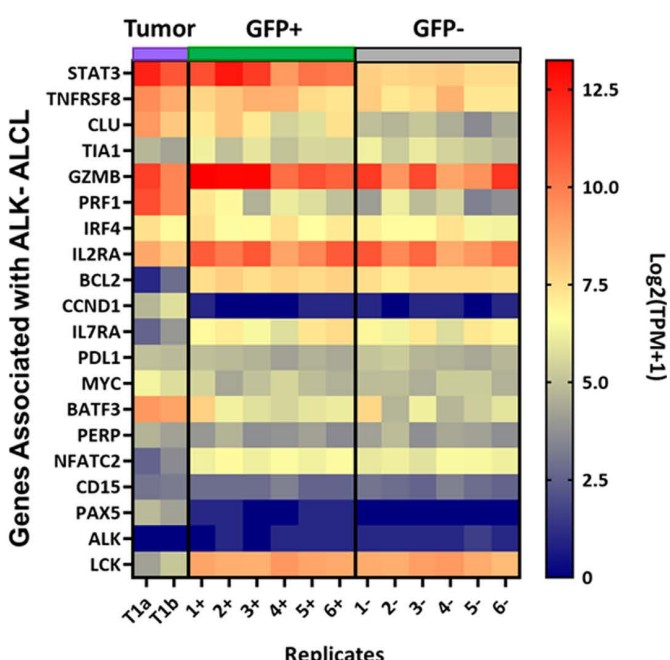

**Fig 7. Similarity of *STAT3* dysregulation and cytotoxic phenotype in GFP-High GFP + Cells to HIV-driven T cell lymphoma.** Bulk RNAseq data from GFP-sorted Donor 4 day 126 samples were compared to data from HIV+ high-grade T cell lymphomas [17] using DESeq2. A) Principal component analysis. B) Heatmap of differentially expressed genes of GFP-High replicates compared to GFP-Low replicates from Fig 3 and high-grade T cell lymphoma. Not all differentially expressed genes are shown. Genes associated with STAT3 signature and cytotoxic phenotype are highlighted. C) Bulk RNA-seq data from Donor 4 replicates and high-grade T cell lymphomas compared to ALK- anaplastic large cell lymphoma related genes. Gene list generated from [31].

The specific loss of infected cells may have been due to modest toxicity of the expressed gfp [49], viral gene products [48], such as prematurely activated protease expressed by the *pro* gene of the vector, or deleterious effects of provirus integration in expressed genes (Fig 2B). Integration in expressed genes is a well-known property of HIV [38,80]. That the decline in fitness of infected cells was accompanied by a specific enrichment of proviruses integrated in intergenic regions, consistent with a deleterious effect on fitness of proviruses integrated into genes, an effect which has also been observed in vivo [38].

The increased *STAT3* expression driven by the HIV provirus shown by these replicates implies that HIV insertional mutagenesis of *STAT3* was associated with clonal expansion or selective survival of these infected cells over time. This result complements our laboratory's previous findings that HIV-driven chimeric transcripts of *STAT3* support clonal

expansion *in vitro* [43]. Although the activating proviruses described in this study were largely in *STAT3* Intron 3, most of the expanded clones in the GFP-High replicates in the current study had proviruses within intron 1, more closely resembling the findings of HIV integration within HIV + T cell lymphomas [17]. We have been unable to identify an experimental basis for the difference between the two studies. Nonetheless, the present study has provided more evidence for proviral insertional mutagenesis of proviral driven *STAT3* as a contributing factor to malignant T cell transformation. Additionally, we have demonstrated increased STAT3 activity in the GFP-High replicates by showing increased phosphorylation of STAT3 at Y705 and increased expression of downstream targets of STAT3 in replicates 2 and 3. The increase of STAT3 abundance by insertional mutagenesis may allow for increased phosphorylation after anti-CD3/CD28 stimulation [70], thus leading to an increase in STAT3-related gene signature in replicates 2 and 3.

The heterogeneity of outcomes of HIV infection and integration-specific activation of *STAT3* among the 3 GFP+ replicates 2, 3, and 4, as well as the absence of such an outcome in the other three replicates, implies that some other genetic or epigenetic difference may also be playing a role in the selective growth and survival of the infected replicate cultures. Although differences in genomic nucleotide sequence or chromatin structure might contribute to differences in outcome, the most likely – and easiest to detect – differences would be in the presence of additional proviruses affecting expression of genes outside of *STAT3*. At the multiplicity of infection used, ~ 2 proviruses/cell, cells with multiple proviruses would not be rare – about 69% of cells with one provirus would be expected to have at least one more. It is noteworthy that clonally expanded infected cells in all three of the high GFP replicate cultures with a provirus driving *STAT3* expression also had at least one additional expanded provirus driving expression of another gene.

The GFP-high replicate 2 cells had the largest number of breakpoints in *STAT3*, and a large majority of the GFP+ cells had increased levels of total and phosphorylated STAT3 protein. Also, qPCR analysis implied that more than 90% of the expanded replicate 2 cells had more than 1 provirus. Together, these results imply that nearly all of the GFP+ cells on day 126 carried a provirus in *STAT3* and a second, antisense-oriented provirus in *CKAP2L*. Insertional mutagenesis at this site of integration led to a high level of expression of an RNA antisense to a region of the *CKAP2L* transcript encompassing exons 1–3 and flanking introns but had no obvious effect on levels of *CKAP2L* RNA expression (Fig 6E). Interestingly, overexpression of the cytoskeletal-associated CKAP2L protein is associated with several solid-tissue cancers, including prostate [81]. Further experimentation will be required to see if there are any effects of the antisense RNA on its translation.

With an average of about two proviruses per GFP+ cell, replicates 3 and 4 showed smaller populations of STAT3 over-expressing cells, possibly suggesting that the proviruses integrated within other genes may have independently contributed to persistence of these cells. In replicates 3 and 4, in addition to *STAT3*, *CCDC77* (rep 3) and *DCUN1D4* (rep 4), are genes with overexpressed in clonally amplified cells in GFP-High replicates due to a provirus in the same transcriptional orientation. Unfortunately, these genes are poorly understood. There is no useful literature on *CCDC77* and all that is published on *DCUN1D4* is that it is an enzymatically defective paralog of *DCUN1D1,* which plays a role in protein neddylation. One paper has identified circular *DCUN1D4* RNA associated with tumor metastasis suppression [82]. Expression vector-directed overexpression of *the* transcripts identified in Fig 6C could provide basic insight into the role of these genes in T cell biology, in addition to the role of clonal expansion in the context of HIV proviral integration in *STAT3*.

Clonally expanded replicate 4 GFP+ cells also contained a provirus driving expression of an RNA antisense to exons 1 and 2 of *GSDMD*, encoding gasdermin D, and overlapping *LOC100310756,* a long noncoding RNA of unknown function. As with *CKAP2L* in replicate 2 (Fig 6D and 6E), there was significant change in anti-sense expression of part of the *GSDMD* gene downstream of the provirus which had no obvious effect on mRNA expression but could have some effect on translation of the gene product (Fig 6E). While clonal GFP/HIV+ replicate 4 cells showed resistance to HIV protease mediated cell death (S6 Fig), further studies would be needed to characterize how these proviral integrations are affecting this pathway. In all these cases, it is unclear whether or how the dysregulation due to the provirus might have affected the growth or survival of the infected cell.

Our data showed an increase in cytotoxic T cell related factors such as granzymes, perforin and genes associated with secretion of these cytotoxic proteins. Findings from recent work characterizing both persistent latently infected CD4+T cells that comprise the HIV reservoir and HIV+T cell lymphomas imply that expression of cytotoxic memory T cell related genes can contribute to cell survival through immune evasion [17,41,42]. As also noted in HIV+T cell lymphomas [17], there is upregulation of *GZMB,* likely driven by the increase in activated STAT3, a hallmark feature of cytotoxic and effector T cells, which we also observed in the GFP-High replicates. The increase in *GZMB* expression was most evident in replicate 2, which also demonstrated the largest increase in STAT3 activation. *GZMB+* cytotoxic CD4+T cells could evade immune responses through killing of nearby immune cells or by being resistant to the effects of other cytotoxic immune cell functions [41,42].

While our culture model lacked specific cytotoxic response to HIV-infected or cancer cells, the HIV+ persistent cells expressed a cytotoxic phenotype which could have killed nearby uninfected cells, as GZMB induces apoptosis through caspase activation. The cytotoxic HIV/GFP+ cells also expressed cell survival genes due to insertional mutagenesis of *STAT3.* Previous studies [62–65] have shown that cells with increased expression of *SOCS3* and *SGK1* are more resistant to Caspase 3- and 7- mediated apoptosis and thus more likely to evade the effects of granzyme B. This cellular phenotype combination of a cytotoxic CD4+T cell and insertional mutagenesis of *STAT3* models a survival advantage, which could be a major contributor to T cell transformation.

As with other retroviral oncogenesis models, insertional activation of STAT3 expression may be rate-limiting first step, with additional mutations likely to be necessary for full transformation of CD4+T cells. While our cell culture model in some ways resembles previously characterized HIV+ lymphomas, the cells in culture are not transformed. Although they can outgrow other cells and clonally expand to become the dominant population, they eventually die off or stop expanding. Therefore, this in vitro transduction and outgrowth system could provide a model to test which additional events need to occur for oncogenesis. For example, the ALK- lymphomas described in Mellors *et. al.*, had increased LCK or HCK activity due to HIV proviral integration, providing kinases that likely contribute to STAT3 activation. Additionally, two HIV+ high-grade T cell lymphomas also had a somatic mutation in *STAT3*, which could have influenced STAT3 activity contributing to oncogenesis. In HIV-negative T cell lymphomas, translocation of genes, such as *DUSP22*, in combination with a gain of function mutation in *JAK1* or *STAT3* can lead to transformation [31]. Therefore, the in vitro model presented in this paper could be used to understand which mutations, in *STAT3*, *LCK* or other oncogenes, would need to occur in concert with HIV insertional mutagenesis events to drive lymphoma generation.

While our in vitro results provide a model for HIV-driven oncogenesis, it is still only a model and has some limitations. This in vitro system does not experience the same immune signals and pressures as cells found in vivo. We do not provide secondary signals for T cell differentiation therefore HIV DNA integration may influence cells differently based on their subtype or phenotype [37]. For example, we see an increase in T cell memory phenotype with replicates containing provirus integrated within *STAT3* (S5B and S5C Fig), but this increase may not be representative of the in vivo situation. Additionally, we used a replication incompetent Gfp-reporter HIV vector, which is unable to express cytopathic HIV proteins such as Env, Vif and Vpr [48,83], which could influence clonal selection and whose integration sites might provide a survival advantage. Additionally, Nef has been recently described as have a protective role in cell survival [84], which could also affect the influence of specific proviral insertional mutagenesis events.

While a few reports in recent years clearly demonstrate the importance of HIV LTR-driven activation of *STAT3* expression in some EBV-negative AIDS-associated B and T-cell lymphomas, difficulties in obtaining suitable samples have hampered our efforts to assess the importance of this mechanism to the overall burden of these lethal malignancies. Whatever its frequency, the LTR insertional activation mechanism suggests a possible novel therapeutic approach [45], based on inhibiting the effects of insertional mutagenesis by silencing provirus-driven transcription. Currently, some HIV cure-oriented research has focused on HIV proviral silencing by "block-and-lock" strategies [85]. A small molecule Tat inhibitor, didehydro-corticostatin A, (dCA), has been identified as a silencer of HIV-LTR driven expression [86–88]. Our in vitro model

could be used to provide insight into the role of insertional mutagenesis in cell survival and oncogenesis and to test the ability of Tat blocking agents, like dCA, to silence the provirus driving *STAT3* expression in vivo. We have been unsuccessful in obtaining dCA, but our model would provide a system in which be tested for its potential as therapeutic for HIV+ lymphomas.

Our in vitro model of HIV+ cell persistence could similarly be used to understand the role of insertional mutagenesis in T cell persistence and oncogenesis associated with HIV-derived ("lentiviral") vector therapies. Previous research has found that insertional mutagenesis by such vectors used in CAR-T therapy was associated with increased survival and proliferation of the vector infected cells [89]. In one example, insertionally inactivated expression of *TET2*, which encodes an important regulator of DNA methylation, was observed in clonally expanded vector-modified T cells making up close to 100% of the total CAR + T cell population at day 121 post infusion [89]. Insertional mutagenesis in this instance led to clonal expansion, but not to lymphoma. Based on some other cases, the FDA has issued a warning that currently approved CAR-T cell therapies using modified HIV vectors to treat B cell malignancies can lead to an increased risk of developing secondary T cell lymphomas [10,11,90]. In most cases, the cause for this increase in lymphoma development has yet to be determined [9,11,91] but integration of the gene therapy vector seems to play a role in a handful of these secondary malignancies [12–15]. Our long-term T cell culture protocol, perhaps in conjunction with animal studies, could be useful to determine which integration sites associated with of a CAR-T vector could provide a secondary mutation likely to lead to oncogenic transformation of the modified T cells.

The persistent HIV/GFP + T cells we describe here provide a model of HIV+ cells likely to be susceptible to oncogenesis in at least some PWH. The upregulation of anti-apoptotic factors and the expression of cytotoxic proteins that promote immune evasion could provide a survival advantage to HIV + CD4 + T cells in vivo that in combination with additional mutations could ultimately lead to transformation and lymphoma development. Our study provides a model for future investigations of the role that HIV and modified HIV vector proviral insertional mutagenesis play in T cell oncogenesis.

## Materials and methods

### Primary human CD4 + T cell in vitro culture

Primary - CD45RA + CD4 + T cells from normal human donors 3, 4 and 5 were obtained from STEMCELL Technologies, inc. (Cells from donors 1 and 2 were described in [43]). Our study did not include any live participants. Cells were cultured in RPMI 1640 (Gibco) supplemented with 10% Fetal Bovine Serum (Gibco) and 1% Pen/Strep (Gibco) and recombinant IL-2 (30 U/mL) (STEMCELL Technologies). Cells were stimulated as needed using anti-CD3/CD28 stimulation beads, Dynabeads™ Human T-ActivatorCD4/CD8 for T Cell Expansion and Activation, (Gibco, Cat#11131D) for 3–4 days and magnetic Dynabeads™ removed. T cells were maintained at about $5x10^5$ per ml of medium. Recombinant IL-2 was added every 3 days (30 U/mL) and T cells were stimulated with anti-CD3/CD28 beads whenever the count started to decrease (approximately every 10–14days) (Fig 1A). when the cell count exceeded $5x10^5$ cells/mL new medium was added but old medium was not discarded, unless cell counts began to drop below $5x10^5$ cells/mL.

### HIV transduction

After initial stimulation with anti-CD3/CD28 beads, primary T cells were infected with a replication incompetent NL4–3 ΔEnv ΔNef EGFP vector [47], with EGFP in place of Nef. The vector was pseudotyped with VSV-G. Spinoculation was performed at 800xg for 90minutes at 37°C with 1μg/mL of polybrene to increase transduction efficiency. Cells were infected at a MOI of 1 *gfp*-transducing unit per cell (TU/cell). Virus-containing medium was removed immediately after spinoculation and replaced with fresh IL-2-containing RPMI media.

### GFP nomenclature

Throughout this paper the nomenclature for the gfp reporter varies to follow standard naming of retroviral gene and protein and to differentiate genotypes from phenotypes. The use of *gfp* refers to the gene within the HIV vector, Gfp will refer

to the protein. Finally, GFP +, GFP- and HIV/GFP+ will refer to the phenotype of the cells that either express or do not express Gfp.

### Flow cytometry and FACS

Primary T cells were analyzed for cell viability using propidium iodide (PI) (Invitrogen, Cat#: P3566) or Zombie Violet (Biolegend, Cat# 423114) staining. GFP phenotype was used to identify cells with active HIV expression. Cell viability and Gfp expression were assessed by flow c cytometry using an LSRII (BD Biosciences) or Attune NxT (Thermo Fisher) analyzer. FACS was used to sort live cells into GFP+ and GFP- using a FACSAria II Cell Sorter (BD Biosciences) and Bigfoot Spectral Cell Sorter (ThermoFisher). To assess intracellular protein expression, T cells were fixed and permeabilized using Cyto-Fast Fix/Perm Buffer Set (Biolegend, Cat#: 426803), then stained with monoclonal antibodies for STAT3 (Abcam, Cat#: ab300101) and GZMB (Biolegend, Cat#: 372219). Isotype control for STAT3 staining (Abcam, Cat#: ab199093) and isotype control for GZMB (Biolegend, Cat#: 400130). For pSTAT3 staining, T cells were fixed and permeabilized using the True-Phos Perm Buffer Set (Biolegend, Cat#: 425401) then stained using monoclonal antibody for pSTAT3 (Y705) (Biolegend, Cat#: 651008) and isotype control (Biolegend, Cat#: 400130). Protein expression was assessed as above. For analysis of T cell phenotype, T cells were stained with antibodies for CD45RO (Biolegend, Cat#: 304215) and CCR7 (Biolegend, Cat#: 353203) and isotype controls (Biolegend, Cat#: 400331 and 400212) used to set population gates.

### Provirus quantification (qPCR)

Genomic DNA was extracted and prepared using DNeasy Blood & Tissue Kit (Qiagen, Cat#: 69504). HIV proviral LTR sequence primers (Fwd: ATGCTGCATATAAGCAGCTGC, Rev: GAGGGATCTCTAGTTACCAGAG) were used in comparison to sequence primers for the single-copy *CCR5* gene (Fwd: CAAAAAGAAGGTCTTCATTACACC, Rev: CCTGTGCCTCTTCTTCTCATTTCG) to quantify proviruses per cell. A standard curve of HIV provirus per cell was used based on mass calculations of the HIV vector plasmid diluted in cellular DNA (Raji cells) (S1 Fig). This standard curve was used to determine that there were about 2 proviruses per cell.in DNA from the ACH-2 cell line, which originally contained a single HIV provirus, but has been shown to accumulate more proviral copies during passage [51,92] (Courtesy of the Kearney Lab, NCI) and used to provide a standard curve for quantifying proviral DNA in the donor 4 samples. qPCR reactions were performed using SsoAdvanced Universal SYBR Green Supermix (Bio-Rad, Cat#: 1725274) and read out using aCFX Connect Real-Time System (Bio-Rad).

### Integration site analysis (ISA)

DNA was extracted from cells using a DNeasy Blood & Tissue Kit (Qiagen, Cat#: 69504). Linker-mediated PCR and library preparation followed the Wells *et al* [44] protocol. 1ug of DNA from each sample underwent fragmentation, end repair and linker ligation using the NEB Next Ultra II FS DNA Library Prep Kit for Illumina (New England Biolabs, Cat#: E7805S). Linker-mediated PCR was performed followed by nested PCR which was added in the Illumina adapters for sequencing. The primers used were as described by the Wells *et al* [44] protocol except that they lacked the 8-nuecliotide inline-index sequence. Paired-end sequencing was performed on the Illumina MiSeq V3 and reads of 150 NT in length were obtained.

Using the Illumina adapters, which contained both 5' and 3' indices, samples were demultiplexed. Each sample underwent pre-alignment filtering and trimming, removing sequences using cutadapt [93] to remove linker and HIV LTR sequences. After trimming, sequences less than 20 NT were discarded. Reads were then aligned to the hg19 genome using Hisat2 [94]. Each alignment was annotated and unique sites, and number of breakpoints were determined utilizing python. The ISA Pipeline can be found on GitHub. For visualization and analysis, unique integration sites that were located close to one another were added together to get an estimation of clonality of integration within specific sites.

### RNAseq analysis

RNA was prepared using RNeasy Mini Kit (Qiagen, Cat#: 74104) on both unsorted and sorted cells. A cDNA library was prepared from unsorted cells using TruSeq mRNA library preparation (Illumina) and single-end sequences (150 bp read length were determined using a HiSeq (Illumina) sequencer. Sorted samples had less RNA, therefore libraries were prepped using TruSeq Stranded Total RNA with Ribo-Zero (Illumina) and analyzed using the NovaSeq6000 (Illumina) with a 150 bp single end read length.

### Transcriptome visualization

RNAseq reads were aligned to the hg38 reference genome using hisat2 [94]. The stranded, aligned sequences were used for visualization using the Integrated Genome Viewer (IGV) software [95,96]. IGV coverage plots were color coded based on the orientation of aligned reads.

### Differential gene expression and gene set enrichment analysis

HTseq [97,98] was used to count hg38-aligned reads for each sample. Transcript counts were used to calculate TPM and used for differential gene expression by DESeq2 (56), a well characterized pipeline for normalizing gene counts and determining differentially expressed genes. Genes that had a log2Fold change greater than 0.5 and adjusted Pvalue less than 0.05 were considered differentially expressed.

Gene set enrichment analysis (GSEA) [57] was used on normalized gene counts from DESeq2 output. The normalized counts were compared between the groups. The number of permutations was 1000 and the permutation type was gene_set, as there were less than seven replicates per group. The gene sets were derived from the Molecular Signatures Database (MSigDB). Hallmark gene sets, groups of 50 pre-defined genes whose expression is positively or negatively associated with specific cell processes, were selected for analysis. To determine the significance of a gene set, the false discovery rate (FDR) q-value was used to adjust for the number of permutations, thus accounting for the likelihood that a gene set is significantly enriched given the number of permutations. FDR q-values less than 0.05 were considered significant.

### STAT3 variant comparison

Hg38 aligned RNAseq reads were sorted and indexed. Bcftools [99] was used to identify variants compared to the hg38 consensus at individual bases in the STAT3 gene. A quality score greater than 90 was used to identify *STAT3* variants from donor 4 RNAseq reads in S1 Table.

## Supporting information

**S1 Fig. ACH2 Cellular DNA Contains Two HIV Proviruses per Cell.** Masses of HIV-vector plasmid and cellular genome were calculated to determine an estimate of the number of proviruses per cell. The standard curve used mixtures of plasmid and Raji cell DNA starting at 10 proviruses per cell and diluted to 0.0033 proviruses per cell. X-axis is the Cq value of the LTR primer divided by the host gene CCR5 primer. Y-axis is the Log10 of the dilution. ACH2 cells are indicated by the red dot.
(TIF)

**S2 Fig. HIV proviral integration in STAT3 drives transcripts downstream of integration site.** Bulk RNA-seq of unsorted donor 4 replicates was performed at day 126 post-transduction. Reps 2,3, and 4 were designated as GFP-High (Blue) replicates and Reps 1,5, and 6 as GFP-Low (Red). The ratio of coverage of *STAT3* exon 2, downstream of the integration site, to that of upstream exon 1 in each replicate (see Fig 3B) allowed us to identify proviral driven transcripts in *STAT3*.
(TIF)

**S3 Fig. GFP+ Cells from GFP-High Replicates Have Increased GZMB expression.** Flow cytometry of Donor 4 day 126 replicates was used to analyze GZMB expression. Cells were gated on viability using Zombie Violet. Cells were then gated on GFP expression and Frequency of GZMB positive cells was compared between GFP+ and GFP- cells of each replicate. (TIF)

**S4 Fig. GFP-High GFP+ Cells Show Increase in Cell Cycle Gene Expression.** GFP-High replicates were sorted on GFP expression. Pooled RNA-seq data from the GFP-High replicates 2, 3, and 4 were used for subsequent analyses. A) Gene ontology analysis was performed using DAVID [100 ,101] on differentially expressed genes comparing GFP+ cells to GFP- cells. B) DAVID was used to compare KEGG Pathways. C) Normalized cell counts were used for GSEA to identify gene set enrichments between GFP+ and GFP- cells. FDR q-Value <0.001 for both gene sets. (TIF)

**S5 Fig. GFP/HIV+ Populations with Proviral Integration in *STAT3* display Increased Memory T Cell Phenotype.** A-B) RNAseq data from sorted GFP-High replicates were used to determine expression of genes associated with different T cell subtypes and naïve, memory or effector T cell phenotypes. C) Flow cytometry was performed on late time point GFP-High replicate cells. Cells were gated on GFP expression and stained for CD45RO (PacBlue) and CCR7 (PE) to differentiate T cells from naïve and different memory phenotypes. (TIF)

**S6 Fig. Clonal Replicate 4 HIV/GFP+ Cells are Resistant to Efavirenz-Mediated Cell Death.** Replicate 4 cells at day 31 and 114 time points were assessed for GFP expression over short time periods with different treatment conditions of DMSO, 5µM LPV, or 5µM EFV. Early and late time point replicate 4 cells were assessed for GFP expression over the course of 96hrs and 72 hrs, respectively, with different culture conditions. GFP was normalized to pre-treatment GFP percentage and used to compare the effects of the different conditions. GFP was used as a reporter for HIV+ cells. (TIF)

**S1 Table. No Somatic Mutations in Protein-coding Exons of *STAT3*.** A "call variants" analysis was used to compare STAT3 transcripts from Donor 4 replicates to the hg38 consensus *STAT3* sequence. Individual nucleotide mutations are identified in the table. All mutations occur in untranslated regions of *STAT3*. (TIF)

**S1 Data. Key to Data for Rist et al.** (XLSX)

## Acknowledgments

We would like to thank the current members of the Coffin Lab, Dr. Michael Freeman, and Daniel Murimi-Worstell, for their comments and helpful discussions. We thank former members of the Coffin Lab Dr. Aidan Burn and Dr. John Yoon for helpful discussions about bioinformatic analysis and for previous work contributing to experimental design, respectively. We would like to thank Ann Wiegand from the laboratory of Dr. Mary Kearney (National Cancer Institute) for providing DNA from ACH-2 cells.

## Author contributions

**Conceptualization:** Michael Rist, Machika Kaku, John M. Coffin.

**Data curation:** Michael Rist.

**Formal analysis:** Michael Rist.

**Funding acquisition:** John M. Coffin.

**Investigation:** Michael Rist, Machika Kaku.

**Methodology:** Michael Rist, Machika Kaku.

**Project administration:** John M. Coffin.

**Resources:** John M. Coffin.

**Software:** Michael Rist.

**Supervision:** John M. Coffin.

**Visualization:** Michael Rist.

**Writing – original draft:** Michael Rist.

**Writing – review & editing:** Machika Kaku, John M. Coffin.

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
