## [Decision Letter · Decision Letter 0]

PPATHOGENS-D-25-00792

Ex vivo HIV DNA integration in STAT3 drives T cell persistence—A model of HIV-associated T cell lymphoma

PLOS Pathogens

Dear Dr. Coffin,

Thank you for submitting your manuscript to PLOS Pathogens. After careful consideration, we feel that it has merit but does not fully meet PLOS Pathogens's publication criteria as it currently stands. Therefore, we invite you to submit a revised version of the manuscript that addresses the points raised during the review process.

Please submit your revised manuscript within 30 days Jun 28 2025 11:59PM. If you will need more time than this to complete your revisions, please reply to this message or contact the journal office at plospathogens@plos.org. Please include the following items when submitting your revised manuscript:

We look forward to receiving your revised manuscript.

Kind regards,

David T. Evans

Academic Editor

PLOS Pathogens

Richard Koup

Section Editor

PLOS Pathogens

Sumita Bhaduri-McIntosh

Editor-in-Chief

PLOS Pathogens

orcid.org/0000-0003-2946-9497

Michael Malim

Editor-in-Chief

PLOS Pathogens

orcid.org/0000-0002-7699-2064

**Journal Requirements:**

**Reviewers' Comments:**

**Part I - Summary**

Reviewer #1: In this manuscript by Rist et al., the authors used a primary CD4+ T cell model to study the impact of HIV integration and the persistence of HIV-infected cells. The authors found that HIV integration into intron 1 of STAT3 (and in the same orientation) substantially promotes the persistence of infected cells in multiple replicates of primary cells, with gene ontology showing enrichment not only in STAT3 pathways but also in DNA replication pathways. This is an in vitro mechanistic evidence of the author’s prior studies that HIV integration into STAT3 drives T cell lymphoma. Overall, this is a nicely performed study with orthogonal validations, such as quantitative integration site analysis (using DNA shearing breakpoint counts), electrophoresis confirmation of integration, RNAseq demonstrating HIV-driven STAT3 expression and STAT3/proliferation/survival gene expression, and orthogonal validation of measuring phosphorylated STAT3. Overall, this is a elegantly conducted study helping the field understand mechanisms of HIV integration site-dependent proliferation and persistence of the infected cells.

Reviewer #2: In this manuscript, Michael Rist and colleagues used a culture model of primary CD4+ T cells to investigate the impact of HIV integration on cell proliferation and survival. This work significantly expands this group’s previous work on STAT3 insertional activation by HIV. The manuscript is well written, the data presented clearly, and the conclusion are supported by the experimental data. This is a work of high quality, which will be of great interest for the scientific community focused on HIV persistence, T cell biology, and lymphoma pathogenesis.

To further strengthen this manuscript, I offer a few suggestions and point at a couple of minor issues:

1) There is a discrepancy in the location of the integrant in the opposite orientation of GSDMD in replicate 4. In the main text the integrant is in exon 2, but in the table 1 it is reported as being in intron 2. I checked the location chr8:144641746 and it indeed looks like being in intron 2, according to hg19. Please verify and correct accordingly. This is relevant in my next point.

2) I find the impact of the integrant in GSDMD particularly interesting, in light of the recent discoveries surrounding the inflammasome protein CARD8, which can sense HIV protease, leading to inflammasome activation and cell death by pyroptosis. GSDMD is the main effector molecule downstream of the CARD8 inflammasome in CD4+ T cells (in which IL-1beta and IL-18 are not activated, in the absence of other PAMPs/DAMPs). Given that CARD8 might sense HIV protease activity upon latency reversal and trigger cell death, if the integrant in GSDMD negatively affects GasderminD function, it would provide a survival benefit. In the Authors’ culture, there are no NNRTI that would enhance this effect (EFV and RPV cause premature protease activation in the cytoplasm, rather than after viral budding), but some spontaneous protease activity can occur. The Authors should discuss this process and the possibility that affecting GarderminD, GFP+ cells might be spared by some negative selection due to HIV expression. If they still have frozen cells from this replicate, the Authors could repeat the culture with and without 5uM of EFV and see whether the integrant in GSDMD has an even stronger survival advantage.

3) The HIV construct used for their model is envelope and nef defective. I suggest to expand the discussion on the impact of the lack of these two genes, since Env can be cytopathic, and Nef can dampen T cell activation and reduce activation-induce cell death. The latter process has been studied recently in CAR-T cells (Perica et al., 2025), in which Nef expression extends their survival in vivo https://www.nature.com/articles/s41586-025-08657-0.

4) The sentence in lines 157-159 is convoluted. Please simplify the description of the experimental setup.

5) I understand what you mean in line 189: “CCR5 is a single-copy cellular host gene”. But please clarify that it’s still a diploid gene. SRY on the Y chromosome is an example of a true single-copy gene.

6) Could the Authors speculate, in the discussion, which experiments should be attempted in the future to better assess the impact on T cell survival/proliferation of integrants in genes with unclear role in T cells? Such as CCDC77 and DCUN1D4?

Reviewer #3: This interesting manuscript by Rist et al. provides new insights into how HIV may cause T cell lymphomas. Specifically, the paper builds on prior evidence that HIV integration into the STAT3 gene can drive clonal expansion by demonstrating how the downstream effects of elevated STAT3 provide a survival advantage to infected T cells. To do this they expand upon an ex vivo model of T cell infection where they track clonal expansion of infected T cells. In the process, they provide a potential model for HIV associated T cell lymphoma.

Context: A prior publication from the same group [1] utilized an ex vivo infection model of activated T cells to show there is a selective advantage for expanding T cells with HIV integrated in the sense orientation in the STAT3 gene. Rationale for this advantage includes that HIV-driven transcription led to aberrant splicing between HIV and the host mRNA, producing HIV STAT3 chimeric transcripts. These chimeric transcripts presumably increase STAT3 expression which may promote clonal expansion.

Strengths: In the current study the authors advance their original findings by providing new insights into why HIV integration into STAT3 would lead to T cell expansion. Moreover, the study provides potential mechanisms for why T cells with HIV inserted within STAT3 would have a selective advantage over other T cells.

Novel findings: The authors first directly demonstrate that STAT3 is elevated in clonally expanded T cells containing HIV inserted in the first intron of the STAT3 gene. They then focus on the downstream effects of elevated STAT3 expression. Genes controlled by the STAT3 transcription factor are elevated in these T cell clones. These genes include anti-apoptotic factors SOCS3 and SGK1 as well as genes involved in T cell cytotoxicity such as GZMB, GZMH, NKG7. These findings provide justification for why these expanded clones would have survival advantages. This is discussed within the context of literature showing that GZMB has been associated with HIV persistence in vivo.

Significance: Clonally expanded infected T cells in their model share key attributes with published HIV associated lymphomas. These shared attributes include that they both contain HIV integrations in the sense orientation of STAT3 and they both express GZMB and TNFRSF8 (CD30). Thus, this ex vivo model should be useful for both probing mechanisms that underline HIV persistence and for studying the development of HIV associated T cell lymphoma.

Weakness: While the role of STAT3 in clonal expansion is compelling, it remains possible that other HIV integration sites may also play a role in the impressive T cell expansion occurring in their model.

1. Yoon, J.K., et al., HIV proviral DNA integration can drive T cell growth ex vivo. Proc Natl Acad Sci U S A, 2020. 117(52): p. 32880-32882 .

**Part II – Major Issues: Key Experiments Required for Acceptance**

Reviewer #1: Given that these are primary cells from uninfected donors (in vitro), not patient cells (ex vivo), please change “ex vivo” to “in vitro”.

Reviewer #2: (No Response)

Reviewer #3: Major issues:

1. In my experience the ACH-2 cell line has a higher level of proviral DNA per genome (~4 copies of HIV/ diploid genome) than reported in this study. Notably, a study from Symons et al demonstrated that ACH-2 cell line has approximately ~100 unique integration events [2]. In fact, this study demonstrated that the number of integration sites increases over time in culture due to ongoing replication.

I recommend the authors carefully calibrate the level of integration in this cell line. For example, they could perform limiting dilution PCR on purified DNA from a precisely quantified number of cells to determine the HIV copy number. This is relevant to the conclusions of Figure 6 A. In this Figure, the authors conclude that replicates 2,3, and 4 have between 1 and 1.5 integrations per cell. This conclusion assumes their ACH-2 control cell line has 1 integration per cell on average. However, if the ACH-2 cell line has 2 or more integrations per cell then the average number of integrations per cell in their ex vivo culture is likely higher than 3 per cell. In this scenario, it seems likely that multiple integrations may play a role in whether a clone expands. While the potential role of additional integration sites is acknowledged in the paper, carefully calculating the average number of integrations per cell informs the reader of the likelihood of this confounder.

2. In their model, the frequency of intergenics increases over time from 41% to 57% (assuming that the sites identified by “.”are intergenics, 2911 out of 5807). Prior studies indicated that the majority of integration sites are in genes when assayed 48 hours after infection of a cell line [3] or when detected in vivo[4]. Can the authors discuss their findings in the context of this literature?

3. Given the role of STAT3 in T cell development, it would be interesting to compare the T cell phenotype of the T cell clones with HIV integrated in STAT3. If this is not feasible, a strong surrogate might be to compare the T cell phenotype of the GFP+ and GFP- cells at late time points (~day 150).

2. Symons, J., et al., HIV integration sites in latently infected cell lines: evidence of ongoing replication. Retrovirology, 2017. 14(1): p. 2.

3. Schroder, A.R., et al., HIV-1 integration in the human genome favors active genes and local hotspots. Cell, 2002. 110(4): p. 521-9.

4. Maldarelli, F., et al., HIV latency. Specific HIV integration sites are linked to clonal expansion and persistence of infected cells. Science, 2014. 345(6193): p. 179-83.

**Part III – Minor Issues: Editorial and Data Presentation Modifications**

Reviewer #1: 1. Can the authors add leading edge genes to Figure 3E GSEA – is STAT3 one of the leading edge genes?

2. “Additionally, insertional mutagenesis of STAT3 driven by the HIV proviral LTR was confirmed through detection of spliced HIV-STAT3 chimeric transcripts initiated in the 5’ LTR.” For HIV-driven splicing into host RNA, the authors may want to cite Cesana Nature Comm 2017, Liu Science Translational Medicine 2020, and Christian ML J Immunology 2022.

Reviewer #2: (No Response)

Reviewer #3: None.

PLOS authors have the option to publish the peer review history of their article (what does this mean? ). If published, this will include your full peer review and any attached files.

**Do you want your identity to be public for this peer review?** For information about this choice, including consent withdrawal, please see our Privacy Policy .

Reviewer #1: No

Reviewer #2: **Yes: ** Francesco R. Simonetti

Reviewer #3: No

**Figure resubmission:**
---

## [Editor Report · Decision Letter 1]

Dear Dr. Coffin,

We are pleased to inform you that your manuscript 'In vitro HIV DNA integration in STAT3 drives T cell persistence—A model of HIV-associated T cell lymphoma' has been provisionally accepted for publication in PLOS Pathogens.

Best regards,

David T. Evans

Academic Editor

PLOS Pathogens

Richard Koup

Section Editor

PLOS Pathogens

Sumita Bhaduri-McIntosh

Editor-in-Chief

PLOS Pathogens

orcid.org/0000-0003-2946-9497

Michael Malim

Editor-in-Chief

PLOS Pathogens

orcid.org/0000-0002-7699-2064
---

## [Editor Report · Acceptance letter]

Dear Dr. Coffin,

We are delighted to inform you that your manuscript, "In vitro HIV DNA integration in STAT3 drives T cell persistence—A model of HIV-associated T cell lymphoma," has been formally accepted for publication in PLOS Pathogens.

Best regards,

Sumita Bhaduri-McIntosh

Editor-in-Chief

PLOS Pathogens

orcid.org/0000-0003-2946-9497

Michael Malim

Editor-in-Chief

PLOS Pathogens

orcid.org/0000-0002-7699-2064